# High-throughput single-molecule quantification of individual base stacking energies in nucleic acids

Jibin Abraham Punnoose [1], Kevin J. Thomas [1], Arun Richard Chandrasekaran[1], Javier Vilcapoma[1], Andrew Hayden[1], Kacey Kilpatrick [1,2], Sweta Vangaveti [1], Alan Chen [1,2], Thomas Banco[1] & Ken Halvorsen [1] ✉

Base stacking interactions between adjacent bases in DNA and RNA are important for many biological processes and in biotechnology applications. Previous work has estimated stacking energies between pairs of bases, but contributions of individual bases has remained unknown. Here, we use a Centrifuge Force Microscope for high-throughput single molecule experiments to measure stacking energies between adjacent bases. We found stacking energies strongest between purines (G|A at −2.3 ± 0.2 kcal/mol) and weakest between pyrimidines (C|T at −0.5 ± 0.1 kcal/mol). Hybrid stacking with phosphorylated, methylated, and RNA nucleotides had no measurable effect, but a fluorophore modification reduced stacking energy. We experimentally show that base stacking can influence stability of a DNA nanostructure, modulate kinetics of enzymatic ligation, and assess accuracy of force fields in molecular dynamics simulations. Our results provide insights into fundamental DNA interactions that are critical in biology and can inform design in biotechnology applications.

Nucleic acids are remarkable in their ability to efficiently carry genetic information, and for material properties that provide high overall stability and still allow biological manipulation. These features are governed primarily by base pairing between two complementary strands and coaxial base stacking between adjacent bases. Although base pairing is often considered to be dominant, both play important roles in nucleic acid structure and function. An interesting example is a minimal RNA kissing complex, with only 2 canonical base pairs but unusually high mechanical stability (similar to a ~10 bp duplex)[1] attributed largely to base stacking interactions[2,3]. Indeed, base stacking is critical to biological processes including DNA replication[4,5], RNA polymerization[6], and formation and management of G-quadruplexes in telomeres[7,8]. Base stacking is also thought to be critical for supramolecular assembly of nucleobases in pre-biotic RNA as part of the RNA world hypothesis[9,10]. Stacking also affects drug development, since small molecule intercalators targeting DNA or RNA rely on

stacking interactions to disrupt a multitude of diverse diseases including cancers, viral infections, Myotonic dystrophy, and Parkinson's disease[11–13]. In biotechnology, synthetic base analogs such as LNA[14], universal bases[15], and size expanded bases[16] partly rely on modified base stacking interactions. The formation of synthetic DNA nanostructures can rely heavily on base stacking, including DNA polyhedra[17], DNA crystals[18], and liquid crystals[19], with some designs assembling using only blunt end stacking interactions[20,21].

Measuring stacking energy between adjacent bases in a helix is challenging due to the small energies, the difficulty in disentangling base pairing and base stacking contributions, and experimental limitations. Early studies used thermal melting spectrophotometry with different terminal overhanging ends to resolve these effects[22,23]. More recent experimental studies of stacking interactions used polyacrylamide gel electrophoresis (PAGE) assays of nicked dsDNA to quantify pairs of stacking interactions[24,25], or optical tweezers to

[1]The RNA Institute, University at Albany, State University of New York, Albany, NY 12222, USA. [2]Department of Chemistry, University at Albany, State University of New York, Albany, NY 12222, USA. ✉e-mail: khalvorsen@albany.edu

monitor binding and unbinding of DNA nanobeams with terminal stacking interactions[26]. These studies have made immense contributions to our knowledge, but their designs and experimental constraints precluded the measurement of base stacking between two individual bases rather than pairs of bases. This has prevented knowing stacking energies between A and C for example, because in the context of a duplex this would be necessarily paired with stacking of the hybridized bases (T and G in this example). In these cases, a single energy value obscures the relative contributions of two or more interaction energies that cannot readily be deconvolved. One recent exception determined the energy of an individual A|G base stack as an application of an imaging technique with magnetic tweezers, but did not systematically investigate base stacking energies[27]. Overall, the lack of data on individual base stacking interactions can limit informed design in biotechnology and synthetic biology where short, engineered contacts are formed between various DNA or RNA strands.

Here we set out to measure individual base stacking interactions at the single-molecule level. Single-molecule pulling techniques can apply biologically relevant picoNewton-level forces to individual molecules, and have been indispensable for the study of biomolecules including folding dynamics and mechanisms of biomolecular interaction[28]. Force is a useful perturbation to compare bond strengths, enabling faster dissociation while still allowing quantification of solution (force-free) behavior. Common single-molecule methods that apply force include optical and magnetic tweezers and atomic force microscopy (AFM). We expanded the single-molecule toolkit with the development of the Centrifuge Force Microscope (CFM), a high-throughput technique that combines centrifugation and microscopy to enable many single-molecule force-clamp experiments in parallel[29]. We have made several iterations to improve the technique, notably enabling single-molecule manipulation with a benchtop centrifuge[30–32], and other groups have advanced the technique as well[33,34]. The high throughput nature of the CFM makes it well suited to collect data from thousands of pulling experiments for a comprehensive assessment of individual base stacking interactions.

Combining the high-throughput CFM with engineered DNA constructs, we quantified individual base-stacking energies of 10 unique base combinations ranging from $-2.3 \pm 0.2$ kcal/mol (G|A stack) to $-0.5 \pm 0.1$ kcal/mol (C|T stack). Stacking energy was not measurably affected by phosphorylation, methylation, or substitution by an RNA nucleotide, but was reduced by a bulky fluorophore modification. Applying our results, we used base stacking to alter the structural stability of a DNA tetrahedron and to change the kinetics of an enzymatic ligation reaction. We also show that our results can be used to evaluate accuracy and potentially improve force fields in MD simulations. Our work provides a comprehensive picture of individual base stacking interactions, as well as concrete examples of how such knowledge can be applied.

## Results

### Premise of experimental design

Base stacking interactions (Fig. 1a) are relatively weak on the order of ~1 kcal/mol, making the measurement of individual stacking interactions challenging. To address this, we considered the design of two duplexes that are weakly held together by identical base pairing but differ by presence or absence of a terminal base stack (Fig. 1b). In the simplest model, this terminal base stack strengthens the interaction and lowers the energy of the bound state (Fig. 1c). The minimal disturbance of the interacting region in the duplex suggests that the transition state should not be appreciably disturbed. This assertion is also supported by previous work that finds only a weak dependence of the transition state to the length of the duplex[35]. The application of external force shifts the process out of equilibrium, allowing only the bound to unbound transition. Measurement of dissociation kinetics can then be used to determine the effect of a single terminal base stack (Fig. 1d). This design allows for flexibility in the overall experimental time scale by control of both the design of the base pairs in the central duplex and by the magnitude of the externally applied force. Building from previous work where we resolved the energy difference of a single nucleotide polymorphism[32], we hypothesized that properly designed single-molecule pulling experiments could resolve the solution-based energies of individual base stacking interactions.

To enable high throughput single-molecule pulling experiments, we used a custom-designed CFM. The CFM is essentially a microscope that can be centrifuged, providing a controlled force application to single-molecule tethers, coupled with video microscopy imaging that can track individual tethers during the experiment (Fig. 2a). Using advances in 3D printing, cameras, and wireless communication electronics, we recently integrated the microscope into a bucket of a standard benchtop centrifuge[32] (Fig. 2b). We achieved live streaming of microscopy images during centrifugation by WiFi communication with an external computer that controls both the centrifuge and the camera through custom Labview software (Fig. 2c). During a typical experiment, we observe tens to hundreds of tethered microspheres in a full field of view at ×40 magnification at a rate of 1 frame per second (Fig. 2d). As the centrifuge spins, force is applied to a DNA construct tethered between a glass slide and microspheres, forcing dissociation of the duplex over time and causing the microspheres to disappear from view (Fig. 2e). Each microsphere is monitored to track individual dissociation events (Fig. 2f), which are used to create a dissociation curve that can be used to extract the off rate (Fig. 2g).

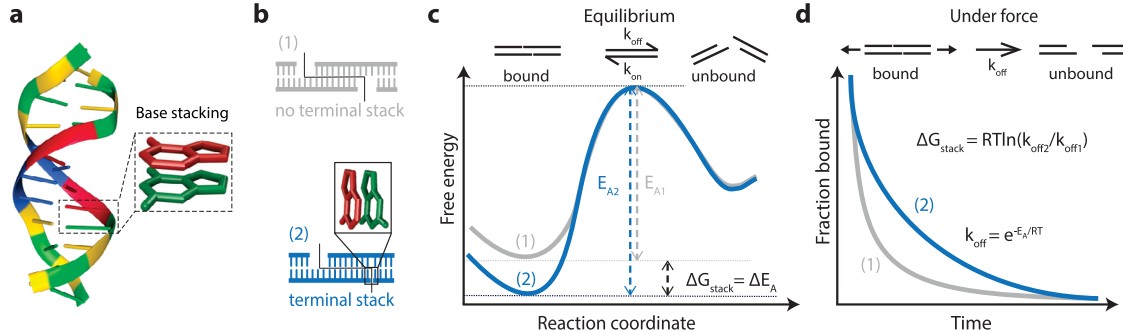

**Fig. 1 | Conceptual overview. a** Model of a DNA duplex[66] with enlarged frame showing stacked adjacent bases. **b** Design of two duplexes differing by a single-base stacking interaction. **c** Free-energy diagram of a DNA duplex with and without a terminal base stack. The base stack primarily increases the activation energy from $E_{A1}$ to $E_{A2}$, with the difference representing the free energy of the single terminal base stack, $\Delta G_{stack}$. **d** External force lowers the activation barriers and prevents rebinding, causing exponential dissociation that can be experimentally measured and used to calculate stacking free energy.

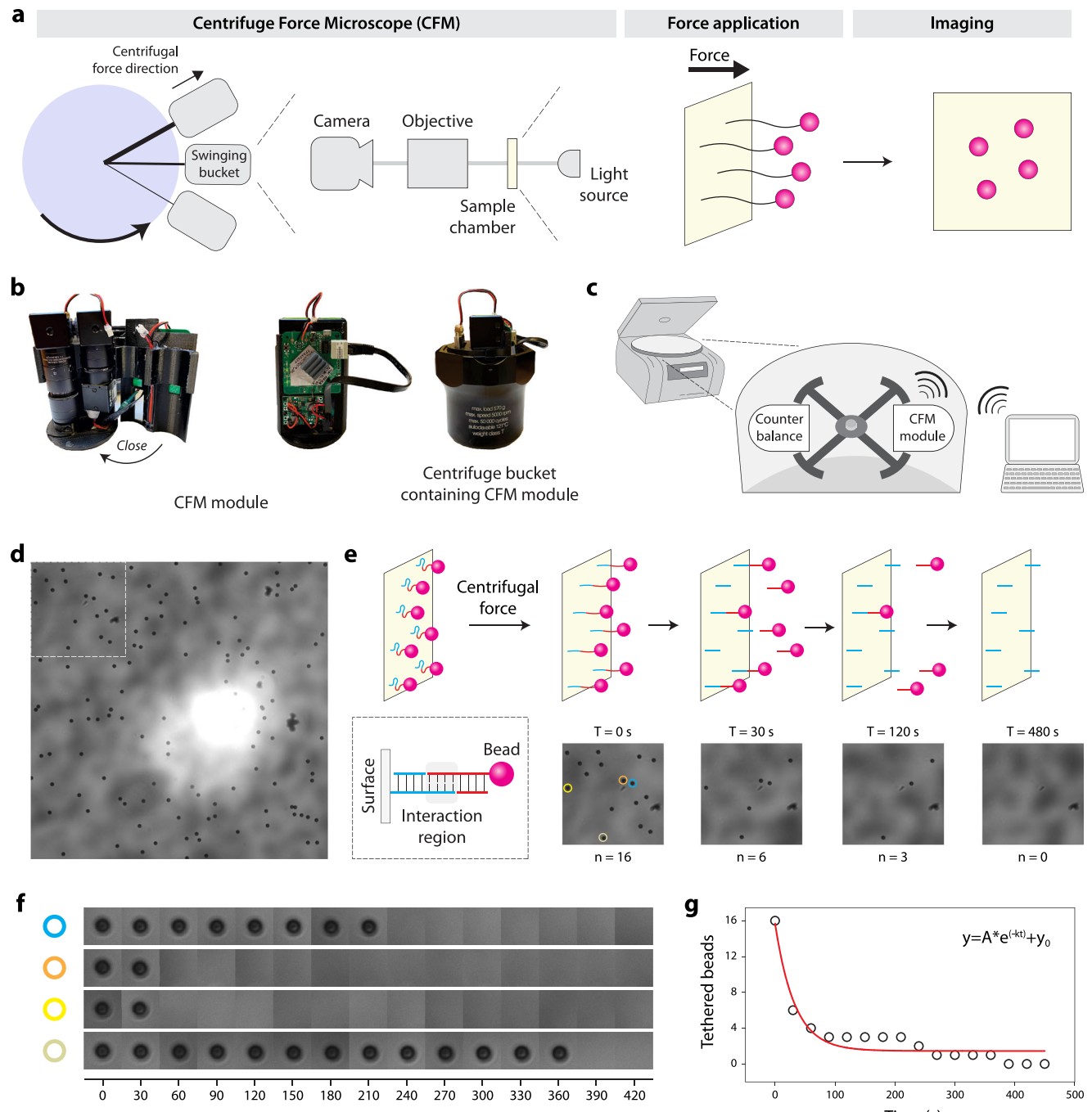

**Fig. 2 | Concept of the centrifuge force microscope (CFM) and force clamp assay. a** The CFM is comprised of a video microscope that is centrifuged. Centrifugal force is applied to tethered microspheres and aligns with the imaging pathway to give a headon view of microspheres. **b** Images of the custom CFM module show the compact central optics, a clamshell style 3D printed housing, and supporting electronics, which fit inside a centrifuge bucket. **c** The CFM module operates in a benchtop centrifuge, which is controlled by an external computer that receives a live video stream by WiFi. **d** A sample microscopy image of ~100 tethered beads at a ~×40 magnification among thousands of such frames. **e** Concept and partial-frame images of tether dissociation observed in the force clamp assay. As the weak central duplex dissociates, tethered beads fall out of focus and disappear from view. **f** Custom MATLAB software tracks tethered beads over time and records dissociation times. Four examples shown with different colors correspond to a subset of beads in panel (**e**). **g** Decay plot obtained from the dissociation time analysis of the 16 tethers in sub-frame (**e**). The red line is a single-exponential fit to extract off rate.

## Experimental measurement of single base stacking energies

As a first test, we confirmed that kinetic differences were measurable between DNA constructs varying by a single base stacking interaction. We designed and created three DNA constructs with identical short central duplexes (8 bp) but varying terminal base stacking interactions (Fig. 3a and Supplementary Fig. 1). In the control construct, a 3 nt poly-T spacer was used to eliminate terminal base stacking completely.

Unlike most previous designs that look at pairs of base stacks or groups of pairs, this design isolates the contribution of a single base stacking interaction between two individual bases. We adopt a notation of X|Y to indicate a stacking interaction between bases X and Y read in the 5′ to 3′ direction, with X residing on the 3′ end of one strand and Y on the 5′ of another. Constructs were created by self-assembly of the 7249 nt M13 genomic ssDNA with complementary tiling

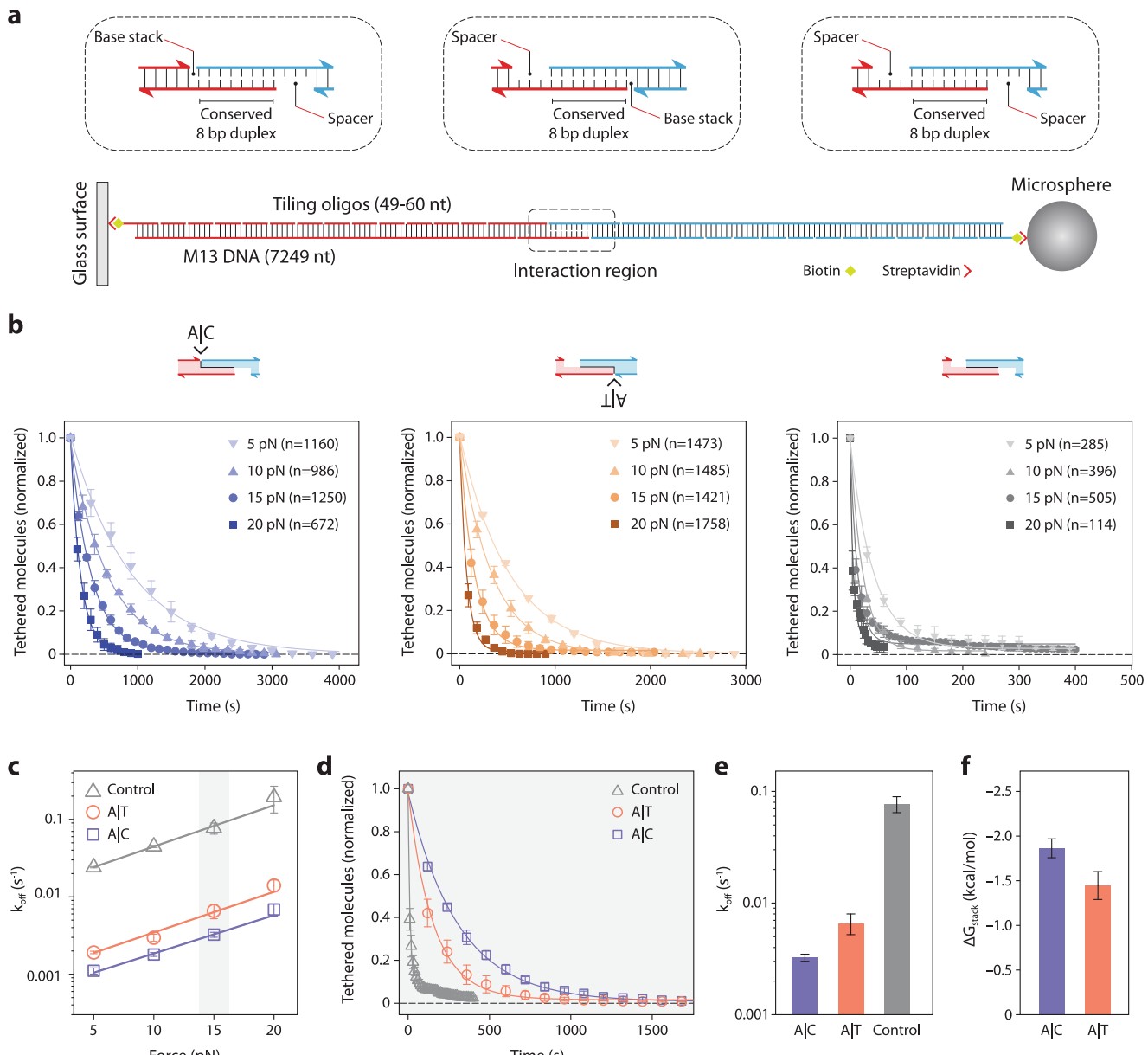

**Fig. 3 | Experimental measurement of single base stacking energies. a** A weak central 8 bp duplex is designed to be flanked by a terminal base stack or no base stacks. The central interaction is formed between two DNA handles attached to a glass slide and a microsphere through biotin-streptavidin interaction. **b** Raw data and single-exponential fits obtained for the A|C, A|T and control constructs at forces from 5 to 20 pN. **c** Force-dependent off-rates fit with the Bell-Evans model (solid lines) to determine thermal off-rate. Force scales determined as $8.8 \pm 0.7$ pN (A|C), $8.3 \pm 1.0$ (A|T), and $8.1 \pm 0.2$ (control), corresponding to transition state distances of $0.47 \pm 0.04$ nm, $0.49 \pm 0.05$ nm, and $0.51 \pm 0.01$ nm, respectively. **d** Analysis of the three constructs at 15 pN shows clear differences in dissociation, fit with exponential decay curves to yield off-rates (**e**), from which $\Delta G_{stack}$ is calculated (**f**). Data in (**b**–**e**) presented as mean values +/− standard deviation from three independent data sets (shown in Supplementary Figs. 2–4). Data in (**f**) calculated from mean values +/− propagated errors from (**e**). *n* represents the number of individual molecular tethers.

oligonucleotides, similar to our previous work with DNA nanoswitches[36,37]. The oligonucleotides tile along the length to make double-stranded DNA, to provide a terminal double biotin for coupling to surfaces, and to provide "programmable" overhanging ends comprising half of the central duplex (sequences in Supplementary Table 1). For the experiment, two pairing DNA constructs were attached separately by biotin-streptavidin interactions to the microspheres and the cover glass. The microspheres were briefly allowed to come into contact with the cover glass within the reaction chamber to allow tethers to form before applying force by centrifugation and measuring dissociation.

We probed the duplexes at forces from 5–20 pN to establish force-dependent dissociation rates at room temperature ($21 \pm 1\,°C$). We hypothesized that the characteristic force scale of different constructs should be nearly identical, which would allow us to extract equilibrium energy differences from off-rates obtained at any constant force. We collected data from over 10,000 single-molecule tethers from multiple experiments that ranged from a few minutes to an hour to ensure most or all beads were dissociated (Fig. 3b). The data were well described by single-exponential decays to determine off-rates at different forces, with $R^2$ values typically exceeding 0.99 (Supplementary Figs. 2–4). Some of the smaller data sets ($n < 100$) had $R^2$ values closer to 0.95.

We note that the y-offset was allowed to be greater than zero, since some data had a small percentage of stuck beads which can arise from anomalous tethers. In some cases it can be observed that there is a continued slow decay over a time scale longer than the single exponential, which is also possibly from rare multiple tethers. While such non-idealities appear minor in our data, we acknowledge that these could provide a source of possible bias in our results. Using the Bell-Evans model[38,39], we fit a linear trend to the logarithm of the force-dependent off-rates for single A|C or A|T stacks and the no-stack control (Fig. 3c). We observed that force-dependent off rates were easily distinguishable between the constructs but followed identical slopes. Since the slopes are related to the position of the transition state, the consistent slopes support our model in Fig. 1c. This result confirmed that individual base stacking interactions could be measured with this approach, and that the choice of force should not appreciably affect the calculated values of equilibrium free-energy of stacking. Similar force-independence has been previously noted in the literature[26]. Using the 15 pN force as an example, it is clear that the three measurements are distinctly different (Fig. 3d, e), enabling the calculation of $\Delta G_{stack}$ by the ratio of off rates (Fig. 3f). We also verified consistency in calculated $\Delta G_{stack}$ across force values and found all results overlapping within error estimates (Supplementary Fig. 5). We decided to proceed with a force of 15 pN, enabling centrifuge runs with hundreds of individual single-molecule experiments to complete in the 10–100 min time scale.

Having successfully proven the concept, we aimed to measure base stacking interactions between all four canonical bases (A, G, C, T) in DNA. Neglecting directionality, the four bases give rise to ten unique base stacks T|T, C|T, A|T, G|T, C|C, A|C, G|C, A|A, G|A, and G|G. We designed DNA constructs to isolate the effect of a single base stack for all 10 combinations (Supplementary Fig. 6). To accomplish this with minimal disturbance to the central duplex, we designed the central duplex to have A, C, T, and G as the 4 terminal bases. This design allowed manipulation of the strands to accomplish all of the 10 combinations with only two control constructs (oligonucleotides listed in Supplementary Tables 1–3).

For each construct and control, we ran experiments at 15 pN at room temperature until most beads have dissociated. Each condition was run with at least three experimental replicates, where each run also contained tens to hundreds of individual tethers. We collected and analyzed over 10,000 single molecule tethers to measure the 10 base stacking interactions. From the images of each run, we measured the dissociation time for each molecule, constructed the decay plot, and found the off-rate by fitting with a single-exponential decay (Fig. 4a, b, Supplementary Figs. 7–9). We determined base stacking energies for all ten base stacks, ranging from −2.3 ± 0.2 kcal/mol for G|A (the strongest) to −0.5 ± 0.1 kcal/mol for C|T (the weakest) (Fig. 4c, Table 1). We observed a general trend that stacking energetics follows the order purine-purine > purine-pyrimidine > pyrimidine-pyrimidine. It is interesting to note that the two control constructs had nearly identical off-rates even with a 5′ to 3′ reversal of the central duplex.

## Influence of nucleotide modification on base stacking energy

Various chemical modifications on nucleotides can influence base stacking and base pairing energy thereby affecting the stability of nucleic acid structures[40]. We extended our approach to probe the effect of these types of modifications in comparison with canonical bases. In particular, we chose phosphorylation, methylation, fluorescein (6-FAM), and substitution of deoxyribose to ribose to study their impact on stacking of the A|C base stack (Fig. 5a). We designed modified oligonucleotides and constructed four duplexes with modified A|C stacks and individual no-stacking controls for each modification (Supplementary Fig. 10). Analogous to the regular base stacking experiments, we performed 15 pN force clamps and fit decay plots to obtain the off-rates (Fig. 5b, c, Supplementary Figs. 11–12) used

to calculate the stacking energy of the modified A|C base stack. The control constructs were all found to be consistent within error. We observed that phosphorylation, methylation, and hybrid DNA-RNA stacks are not appreciably different from the regular A|C base stack, while the bulky FAM group reduced the base-stacking energy by 0.8 ± 0.2 kcal/mol (Fig. 5d). These results show that stacking effects of chemical modifications can be measured with our technique, and suggest generally that small modifications are less likely to interfere with stacking under the conditions tested here.

## Base-stacking in biotechnology applications

Elucidation of these base stacking energies can benefit many aspects of biotechnology, which often rely on forming or dynamically controlling short DNA duplexes. These include molecular biology methods such as genetic recombination, polymerase chain reaction, and sequencing, as well as emerging technologies like gene editing, synthetic biology, and DNA nanotechnology. These stacking energies can also help influence molecular simulations, whose accuracy relies on parameters that reflect realistic potentials between the simulated components. Here we show how our results can be used to benefit DNA nanotechnology, enzymatic ligation, and molecular dynamics (MD) simulations.

In DNA nanotechnology, DNA is used as a building block for nanomaterials[41] with applications including drug delivery[42] and sensing[43]. The field relies on forming controlled contacts between short DNA segments. We hypothesized that designing DNA motifs with specific interfacial base stacks could alter the assembly and stability of DNA nanostructures. To test this, we used the DNA tetrahedron as a model system, a widely used structure with biosensing and drug delivery applications[17,44]. The DNA tetrahedron is hierarchically self-assembled from 3-point-star motifs that connect to each other through a pair of 4-nt sticky ends (Fig. 6a). Our control structure, based on a DNA tetrahedron we previously reported[44], contained two pairs of base stacks (G|A and A|T) across two 4-nt sticky end connections. We annealed the DNA tetrahedron and validated self-assembly using non-denaturing PAGE (Fig. 6b). To test the effect of different base stacking interactions, we modified the sequence of the component DNA strands and constructed three other versions of the DNA tetrahedron with one pair of G|A base stacks, one pair of A|T base stacks, or no base-stacks (Fig. 6c). We observed that structures containing both the G|A and A|T bases stacks were best formed, followed by the G|A structure, while the other two were apparently too weak to form stable structures (Fig. 6d, full gels in Supplementary Fig. 14). To confirm that the G|A design was less stable and not just produced in a lower quantity, we tested thermal stability and observed a decrease in the relative stability of the structures with increased temperature, and a clear indication that the G|A structure was unstable at 40 °C while the G|A + A|T structure was still intact (Fig. 6e, Supplementary Fig. 14). These results are consistent with our findings that G|A base stack is stronger than A|T base stack, demonstrate the crucial role of base-stacking in the stability of DNA nanostructures, and show how altered stability of a DNA tetrahedron can be achieved by design of base stacking interactions. Designing DNA nanostructures typically only involves consideration of the base pairing, and this work points to an additional dimension of control and design flexibility.

DNA ligation is a process of enzymatically joining two pieces of DNA, often facilitated by short sticky ends of 1–4 nt that hybridize together. Ligation is a fundamental biological process that is required for DNA repair and replication and is integral to a wide range of biotechnology applications including sequencing, cloning, and diagnostics[45,46]. We hypothesized that modification of interfacial base stacks could alter the kinetics by changing the lifetime of the bound duplex and potentially the final efficiency of enzymatic ligation. To test this hypothesis, we designed and created short duplexes to enable ligation of products with varying sticky ends (Fig. 6f). First, we validated construction of the individual duplexes and the successful

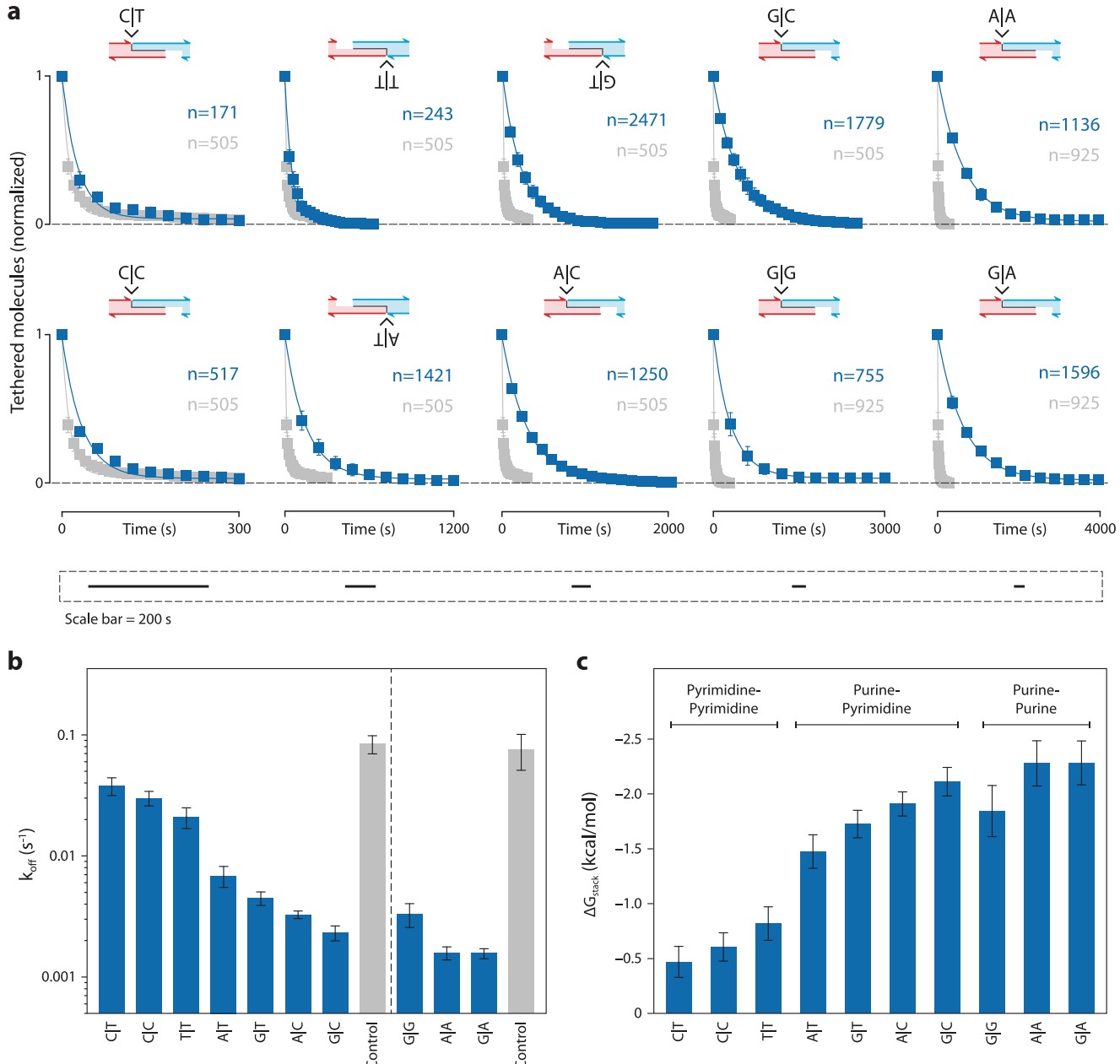

**Fig. 4 | Comprehensive study of DNA base stacking. a** Decay curves and single-exponential fits obtained for unique base stacking combinations (blue) and their controls (gray) at a constant force of 15 pN. **b** Off-rates of DNA tethers containing various base stacks and their corresponding controls. **c** Base stacking energies calculated from panel (**b**). Data in (**a**, **b**) presented as mean values +/− standard deviation from three independent data sets (shown in Supplementary Figs. 7–9). Data in (**c**) presented as calculated value from mean +/− propagated errors from (**b**). $n$ represents the number of individual molecular tethers.

**Table 1 | Individual base stacking energies determined using CFM**

|  | G\|A | A\|A | G\|G | G\|C | A\|C | G\|T | A\|T | T\|T | C\|C | C\|T |
|---|---|---|---|---|---|---|---|---|---|---|
| $\Delta G_{stack}$ (kcal/mol) | −2.3 ± 0.2 | −2.3 ± 0.2 | −1.8 ± 0.2 | −2.1 ± 0.1 | −1.9 ± 0.1 | −1.7 ± 0.1 | −1.5 ± 0.2 | −0.8 ± 0.2 | −0.6 ± 0.1 | −0.5 ± 0.1 |

ligation of the two duplexes (Fig. 6g and Supplementary Fig. 15). Next we investigated the ligation kinetics of four variants, 4 nt and 3 nt sticky ends with either T|A or G|A terminal base stacks (Fig. 6h, i and Supplementary Figs. 16–17). In the 4 nt case, we observed a slight increase in kinetics with the G|A stacks, which was most evident in the first 20 min of the reaction (Fig. 6j). For the 3 nt case, the difference was more striking, with a substantial difference in both the kinetics of ligation as well as the endpoint. The differences can be most clearly seen when looking at the ligated products after an 8 min reaction,

where the trend follows 4 nt G|A > 4 nt T|A > 3 nt G|A > 3 nt T|A (Fig. 6k). Interestingly, the magnitude of the change between T|A and G|A in the 3 nt case is similar to the change between 3 nt and 4 nt in G|A, suggesting that strong base stacking interactions could potentially compensate for weak base pairing in such short duplexes. Building on this idea, we tested whether we could design a 3 bp interaction with strong base stacking that outperforms a 4 bp interaction with weak base stacking. We made a 3 bp design with A|G stacks on both sides and found that it had substantially faster ligation kinetics than the same

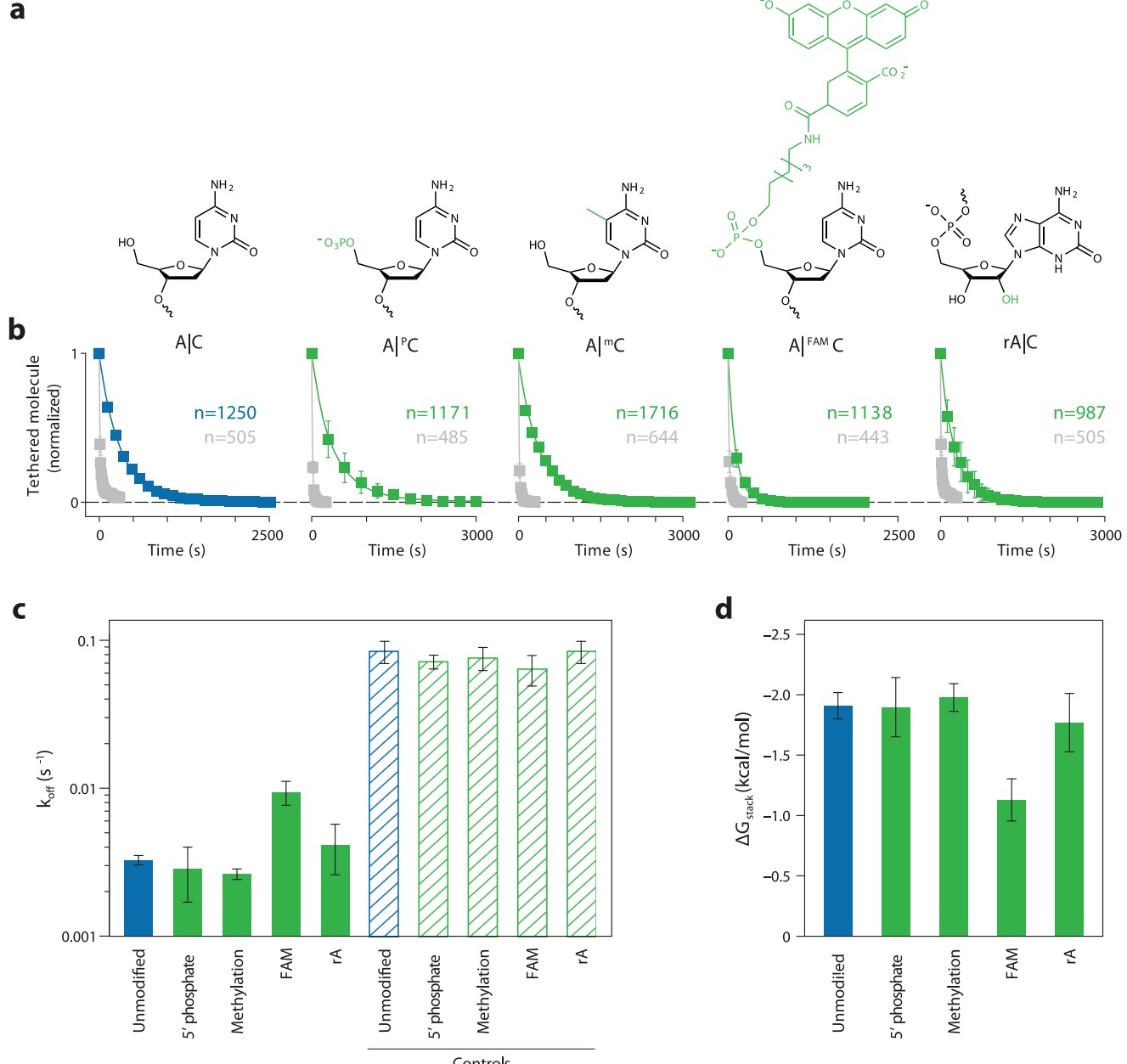

**Fig. 5 | Effect of nucleotide modification on base stacking energy of nucleotides. a** Modifications used in the study including 5' Phosphorylated C, 5-methyl C, 5' FAM modified C, 3' ribose A. **b** Experimental data shows dissociation over time for constructs relative to their controls. **c** Off-rates observed for tethers with modified bases and their controls. **d** Free-energy of stacking calculated from off-rates in (**c**). Data in (**b**, **c**) presented as mean values +/− standard deviation from three independent data sets (shown in Supplementary Figs. 11–12). Data in (**d**) calculated from mean values +/− propagated error from (**c**). *n* represents the number of individual molecular tethers.

sequence with an added A-T pair but with two weaker C|T stacks (Supplementary Fig. 18). These results clearly show how our data can be used in biotechnology applications, presumably for a host of enzymatic interactions that go far beyond ligation.

Molecular dynamics (MD) simulations are a powerful tool to study conformational dynamics of biomolecules including nucleic acids[47]. However, simulations are only as accurate as their underlying empirical energy functions (i.e., force-fields), which must be strategically calibrated against experimental measurements. Force fields for nucleic acids require precise and separate calibration of base-stacking and base-pairing energies for all nucleotide combinations, which is particularly difficult to compare with experimental studies that typically combine these in terms of a "nearest-neighbor" thermodynamic model[48–51]. Previous attempts[52] roughly recalibrated only purine/purine and pyrimidine/pyrimidine based on limited experimental data from dinucleotide stacking measurements, which also present challenges in geometrically defining stacking in the absence of the double helix[53]. Our base stacking results provide accurate thermodynamic measurements of single-base stacking for all possible nucleobase combinations in the context of a double helix, which uniquely enables direct calibration of MD force fields in a sequence-specific manner. Here we used our data to evaluate DNA base stacking for two MD force fields optimized for nucleic acids: Amber-99 Chen-Garcia[52] and parmbsc1[54], the former of which was optimized for RNA, and the latter of which is currently considered the standard force field for MD simulations of DNA (Fig. 6l). To mimic CFM experiments, we simulated

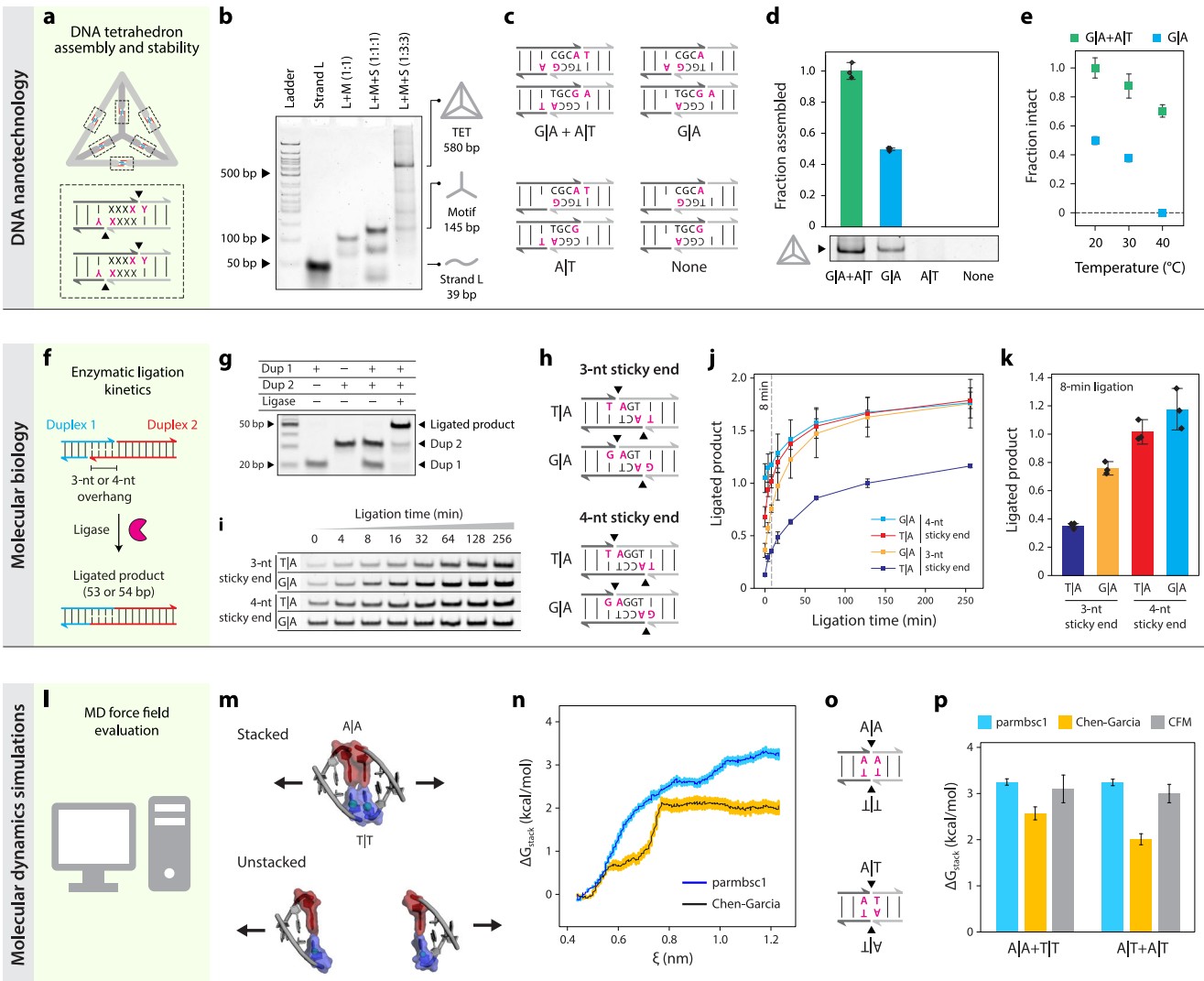

**Fig. 6 | Base-stacking in biotechnology applications. a** A DNA tetrahedron is assembled from four 3-point-star DNA motifs connected on each edge with two pairs of sticky ends. **b** Non-denaturing PAGE analysis shows DNA tetrahedron assembly from oligonucleotide components. (n = 1). **c** Designs of base stacking interactions tested in DNA tetrahedra with conserved sticky end sequences. **d** Assembly of DNA tetrahedra with different base stacks (full gels in Supplementary Fig. 14). **e** Thermal stability of DNA tetrahedra with various base-stacks. **f** Ligation of two DNA duplexes with 3 or 4 base pair sticky ends. **g** Non-denaturing PAGE confirms the ligation of the two DNA duplexes (full gel in Supplemental Fig. 15) (n = 1). **h** Designs of base stacking interactions of sticky ends tested for ligation. **i** Gel images show the increase in band intensity of ligated fragments (full gels in Supplementary Figs. 16–17). **j** Quantified ligation product over time. **k** Ligated product

for different base stacks at 8 min. **l** Molecular dynamics simulations of base stacking interactions with different force fields. **m** Simulation scheme showing two 3 bp duplexes with A|A in red and T|T in blue, in the initial (stacked) and the final (unstacked) confirmation. The pulling force is applied on the C1′ atoms of the T|T stacked pair, orthogonal to the base-pairs. **n** Potential of mean force (PMF) of the A|A-T|T construct as a function of the distance between the pull groups (ξ), for the Amber99-Chen-Garcia[52] and parmbsc1[54] force-fields. **o** Designs of base stacking interactions tested in MD simulation. **p** Free energy of stacking (ΔG$_{stack}$) as calculated from the simulations compared to experimentally determined values (values added from Table 1). Data in (**d**, **e**, **j**, **k**) is presented as mean values +/− standard deviation from triplicate experiments. Data in (**n**, **p**) presented as calculated values +/− standard deviations of the energies sampled at each distance.

two 3-mer duplexes in a solution of ~66,000 water molecules and 177 Na$^+$ ions and 169 Cl$^-$ ions, enclosed in a 10 nm × 20 nm × 10 nm 3D periodic box. Duplexes were stacked end-to-end and pulled apart (Fig. 6m), with potentials computed as a function of the distance between pulling groups to determine the change in energy between the stacked and unstacked configurations (Fig. 6n and Supplementary Fig. 19). We tested two stacking interactions with pairs of either A|A and T|T or A|T and A|T (Fig. 6o) and found that parmbsc1 force field overestimated both stacking interactions while the Amber 99 Chen-Garcia force field underestimated them (Fig. 6p). This work shows that our CFM experimental design can be reliably replicated in a MD simulation and used to evaluate and potentially optimize force field parameters to improve the quantitative accuracy of MD simulations for nucleic acids.

## Discussion

This work provides direct and comprehensive data on base stacking in nucleic acids, while also demonstrating the utility of such detailed knowledge. By employing high-throughput single-molecule experimentation using the CFM combined with programmable design of DNA tethers, we measured tens of thousands of individual interactions and quantified base stacking with an uncertainty of ~0.1 kcal/mol. With such small energies, measuring kinetic rates provides an inherent advantage due to the logarithmic dependence of the energies on kinetics. Single-molecule techniques are a good fit for this, except they typically make just one measurement at a time. The CFM was developed to address limitations of throughput and accessibility in single-molecule research, and this work marks a milestone as one of the first large studies using the CFM. The throughput and accessibility are

evidenced by the ~30,000 single-molecule tethers used in this work, with data collected largely by an undergraduate researcher using a benchtop centrifuge.

Our work provides new data on base stacking, which generally suggest that previous work has underestimated base stacking energies. One striking example is our measurement of −2.3 kcal/mol for a single G|A stack, which is substantially more stable than measured dinucleotide stacks containing both G|A and T|C, which were reported in the −1.0 to −1.6 kcal/mol range[24,26]. When we compared pairs of our measured base stacking values with previously measured dinucleotide stacks, our energies were larger in all cases by multiples ranging from 1.2 to 2.2. It is likely that a mix of different experimental conditions and biases in experimental designs are responsible for these differences. Our experimental approach provides a fairly direct measurement and analysis compared to some other approaches, and utilized PAGE purified oligonucleotides to ensure terminal bases were always present. Previous approaches have required several assumptions to arrive at their results. Deriving energies using stacked/unstacked equilibrium from migration of nicked DNA in urea gels[24] required assumptions about the effect of urea denaturant, the effect of physical forces in gel electrophoresis, and the effect of bending energy on kinked DNA. Deriving energies from measurements of single-molecule in end-stacking of DNA origami tubes[26] required assuming perfect formation and stability of complex DNA origami structures with sharp bends at the measured interfaces, a passive linking tether, and aligned contact across multiple interfaces with force shared evenly among them. It is worth pointing out that failure of most of these assumptions would be expected to result in measuring a base stacking value weaker than the actual value. Notably, one recent paper published with a similar construct design and relatively direct measurement approach found a single A|G base stack energy of −2 kcal/mol[27], consistent within error to our measurement.

Given the comparison with previous results, it is worth exploring assumptions and uncertainties in our study. The work is largely predicated on the model in Fig. 1c, which assumes a 1D energy landscape with a transition state that doesn't change appreciably between constructs. In pulling experiments, the reaction coordinate is constrained to the 1D pulling direction, and a misalignment with the equilibrium reaction coordinate could bias the extrapolation to the zero-force off-rate. Also potentially biasing this extrapolation would be changes in the transition state, which could arise from sequence dependency of the transition state or physical differences such as mechanical forces distorting bond angles[55]. Any deviations from the idealized model in Fig. 1c could cause systematic errors that would affect our end results of base stacking energies. However, we believe these concerns are largely alleviated by (1) our construct design, (2) evidence from past literature, and (3) our own results. We designed constructs to minimize differences; base stacked constructs are structurally identical (only differing by identity of a single base) and control constructs are structurally similar with the insertion of a small ssDNA gap. With this design, we expect extrapolation biases to equally affect all of the base stacked constructs and likely also the non-stacking controls, preserving the stacking energy estimates. Previous literature largely supports this model as well. Pulling on duplexes has been suggested to extrapolate to the equilibrium off-rate[56], even with variations in construct design, suggesting that the 1D model is a reasonable approximation. Further, the transition state only weakly depends on duplex length[35], and helicity and base pair tilt have been shown to not play a major role in duplex shearing[55]. On-rate experiments have shown consistent results between stacked and unstacked configurations similar to ours[27], and have been reported to be sequence-independent for stacking interactions[26]. Together, this body of evidence suggests that small structural variations do not alter the transition state appreciably. Our own experimental data in Fig. 3 supports this empirically by exhibiting parallel slopes in the force dependence of different

constructs, as well as transition state positions that match well with previous literature using different construct designs[56]. The close agreement of our non-equilibrium data with equilibrium measurements under force by Rieu et al.[27] also suggests our approach is valid.

The data presented here may provide new insights into biological processes, inform DNA design in biotechnology, and improve accuracy for molecular modeling. Especially for short sticky ends that are ubiquitous in biotechnology, base stacking can play a surprisingly large role in stability. Our experimental examples of constructing DNA tetrahedra and monitoring DNA ligation provide glimpses of how our data can be used to tune DNA interactions. These results would not be readily predictable from previous data or nearest neighbor approximations, and yet they provide support for both the magnitude and future utility of this base stacking knowledge. While our data was mostly limited to DNA base stacking, our approach can be useful for studying RNA and RNA modifications as well. Our data seems suggestive that RNA base stacking may not be appreciably different from DNA, but with the caveat that our measurement inserted an RNA nucleotide into a DNA duplex, which is known to have a different structural form than RNA duplexes. Further work will be needed to clarify differences in RNA and DNA stacking, and the role of different chemical modifications on base stacking. Our general approach can be adapted to study many variations of nucleotide interactions including those of intercalators under a variety of biologically relevant conditions. It is our hope that this work is appreciated for the unique and translatable approach, for the fundamental information about single base stacking energetics, and for the implications of these results in biotechnology.

## Methods

### Instrumentation

The constant force single-molecule experiments in this study were performed using a custom-built CFM, the details of which were largely reported in a previous study[32]. Briefly, the CFM consist of optics comprising a miniaturized video microscope, and of electronics allowing operation and data transmission, in an assembly that fits within a 400 mL bucket of a Sorvall X1R centrifuge. The optical components consist of a ×40 plan achromatic infinity-corrected objective (Olympus) for microsphere magnification, turning mirrors (Thorlabs) for achieving required path-length and an LED with diffuser as a light source. The electronic components consist of a gigabit Ethernet machine vision camera (FLIR Blackfly Model # BFLY-PGE-50H5M-C) for imaging, a Wi-Fi router (TP-link TL-WR902AC) for wireless data transfer and communication, and a rechargeable lithium-ion battery (Adafruit) with 5 V and 12 V voltage step-up regulators (Pololu). These components were assembled within a 3D printed housing (Ultimaker 3). The CFM module and the centrifuge were controlled using a custom written LabVIEW program.

### Sample preparation

DNA constructs were prepared by hybridizing 124 oligonucleotides (Integrated DNA Technologies) to 7249 nt single-stranded M13mp18 DNA (New England Biolabs). The construction method largely follows our approach for DNA nanoswitch construction[57]. Briefly, the M13 DNA is enzymatically linearized and then incubated with a 10-fold molar excess of backbone oligos 1–122, 150-fold overhang oligo, and 500-fold stacking end oligo (Supplementary Table 2) with an annealing temperature ramp from 90 °C to 20 °C. In this design, the oligo hybridized to the 3′ end of the M13 DNA contains a double biotin on its 5′ end for immobilization to streptavidin-coated glass surface or the bead. The oligo hybridized to the 5′ end of the M13 DNA extends beyond the M13, and provide a platform to anneal an oligo resulting in a 5′ single-stranded overhang (Supplementary Fig. 1). This overhang is used to form 'sticky-ends' for different constructs with various base stacking combinations. The list of all oligos used is given in Supplementary

Table 1 and combination of oligos to make constructs with different terminal bases are given in Supplementary Table 2. A list of reagents used is provided in Supplementary Table 4.

To immobilize DNA constructs to streptavidin-coated microspheres (Thermo Fisher Dynabeads M-270 2.8 µm diameter, catalog # 65306), we used 20 µl of streptavidin microspheres and washed thrice with 50 µl of phosphate-buffered saline containing 0.1% Tween 20 (PBST). Following the washes, the beads solution was brought to a 10 µl volume, and 10 µl of the DNA construct (~500 pM) was added to it and shaken in a vortexer at 1400 rpm for 30 min. The unbound DNA and excess oligos from the construct synthesis was removed by washing the beads thrice with 50 µl PBST and resuspending in 40 µl volume.

The reaction chamber was prepared according to previous work[31]. Briefly, the reaction chamber consists of an 18 mm and a 12 mm circular microscope glass slide (Electron Microscopy Sciences, catalog # 72230-01 & 72222-01) sandwiching two parallel strips of Kapton tape (www.kaptontape.com) creating a channel of ~2 mm between the glass-slides. The glass chamber is assembled on top of a SM1A6 threaded adapter (Thorlabs). Streptavidin (Amresco) was passively adsorbed to the surface by passing 5 µl of streptavidin (0.1 mg/ml) in 1× PBS. After one minute of incubation, the chamber was washed thrice with 50 µl of PBST to remove unbound streptavidin. Next, 5 µl of DNA construct was passed through the channel and incubated for 10 min for the biotin-labeled DNA constructs to bind the streptavidin on the glass surface. The chamber was then washed with PBST to remove unbound constructs and excess oligos from the construct synthesis. DNA-coated microspheres were passed into the chamber and incubated for 10 min to allow hybridization. The chamber was sealed with vacuum grease and then screwed into the CFM optical assembly until the beads are in focus.

### Constant force experiment protocol

The prepared CFM with sample chamber was then loaded into the centrifuge bucket, opposite of a counterbalance with matched mass and center of mass. A custom LabView program was used to control the instrument, including the centrifuge speed, image acquisition rate, and camera parameters such as exposure time. For our 5MP camera the fastest transfer rate was 2 fps, and for these experiments we used 1 fps and saved every fifth frame to reduce data size. This rate was appropriate for our experiments which ran for several minutes to ~2 h. The force generated on the tether is the centrifugal force experienced by the beads $F = m\omega^2 r$, where m is the effective mass of the bead (actual mass minus the mass of buffer displaced), $\omega$ is the angular velocity and $r$ is the distance from the center of the rotor to the chamber (measured at 0.133 m here). The effective mass of beads was determined to be $6.9 \times 10^{-12}$ g for the Dynabeads™ M-270 (www.thermofisher.com) by previous report[23]. The RPM used were 1410, 1221, 997, and 705 (or 296, 222, 148, 74 g) for 20 pN, 15 pN, 10 pN, and 5 pN, respectively. Experiments were run at a constant force (RPM) for times up to 2 h and data was saved as individual lossless images. For subsequent analysis, time zero was defined as the first frame where the final RPM was reached.

### Data analysis

Force-induced dissociation of the DNA tethers were measured using a previously reported MATLAB program[32]. The MATLAB program identifies beads using the "imfindcircles" algorithm. The identified beads were visually checked to ensure beads appeared to be single tethers. Rare anomalous non-spherical beads, closely clustered beads, and dirt or other objects wrongly identified as beads were excluded. In addition, beads out of the typical focus were excluded due to the possibility of multiple tethers. Once beads were identified from an image at the start of the experiment, the software calculated the variance of the image intensity at the bead location for all the frames. When beads dissociate, it is indicated by the sharp drop in variance (i.e., high contrast to low contrast). Multiple drops in variance were rarely observed among analyzed beads due to our pre-screening, and were excluded from analysis due to the possibility of multiple tethers. Raw data of dissociation times were plotted as histograms with bin widths chosen to maintain approximately the same number of bins for each data set even with overall time spans varying by more than one order of magnitude.

The decay rates were plotted in OriginLab and data was fit using single-exponential decay function, $y = y_0 + A \times e^{-kt}$, where $y$ is the fraction of tethers remaining at a given time $t$, $y_0$ is the y-axis offset or the baseline, $A$ is the fraction of tethers at the beginning of the experiment (typically 1) and $k$ is the off-rate for that particular force. Off-rate for any given condition was determined by at least triplicate experiments where individual $k$ values were determined separately for each set of experiment, and data is reported as the mean and standard deviation of the replicates. The base stacking energies were extracted by comparing the off-rates, given by the equation:

$$k_{off} \propto e^{-E_a/RT}. \tag{1}$$

Where, $E_a$ is the activation energy, $R$ is the gas constant and $T$ is the absolute temperature. The off-rates of construct with a particular base-stack can be compared to its control construct without base stack to obtain the difference in activation energy:

$$\frac{k_{off1}}{k_{off2}} = e^{(E_{a2}-E_{a1})/RT}. \tag{2}$$

The difference in activation energy between the two constructs is the energy contribution from the base stack following the assumption that the on-rates are equal between the different constructs[58], which can be isolated using the equation:

$$\Delta E_a = \Delta G_{\text{Base-stack}} = RT\ln\left(\frac{k_{off1}}{k_{off2}}\right). \tag{3}$$

where $k_{off1}$ and $k_{off2}$ are the off-rates of construct with and without the base stack, $E_{a1}$ and $E_{a2}$ are the activation energy barriers for those constructs, respectively, and $\Delta G_{\text{Base-stack}}$ is the stacking energy of the interfacial bases in the non-control construct.

### Assembly and measurement of DNA tetrahedra

DNA tetrahedra were prepared using previously reported methods[43]. Briefly, DNA strands L, M, and S (sequence are shown in Supplementary Table 1) were mixed in 1:3:3 ratio at 30 nM in Tris-Acetic-EDTA-Mg$^{2+}$ (TAE/Mg$^{2+}$) buffer, which contained 40 mM Tris base (pH 8.0), 20 mM acetic acid, 2 mM EDTA, and 12.5 mM magnesium acetate. The DNA solution was slowly cooled down from 95 °C to room temperature over 48 h in a water bath placed in a Styrofoam box. To assemble DNA tetrahedra with different base stacks, the following strand combinations were used:

(1) Tetrahedron with both AG and AT base stacks (control): Strands L, M1, S1
(2) Tetrahedron with AG base stack only: Strands L, M2, S1
(3) Tetrahedron with AT base stack only: Strands L, M1, S2
(4) Tetrahedron with no base stacks: Strands L, M2, S2

DNA tetrahedron assembly was validated using non-denaturing polyacrylamide gel electrophoresis. For gel analysis, 10 µl of the annealed DNA tetrahedron solution was mixed with 1 µl loading dye (containing 50% glycerol and bromophenol blue). 10 µl of this sample was loaded in each gel lane. Gels containing 4% polyacrylamide (29:1 acrylamide/bisacrylamide) were run at 4 °C (100 V, constant voltage) in 1X TAE/Mg$^{2+}$ running buffer. After electrophoresis, the gels were

stained with GelRed (Biotium) and imaged using Bio-Rad Gel Doc XR+. Gel bands were quantified using ImageLab. To analyze the thermal stability of DNA tetrahedra with different base stacking combinations, we incubated the DNA tetrahedra at 30 °C and 40 °C for 1 h. Incubated samples were prepared for gel analysis as described above and tested using 4% PAGE. We quantified the band corresponding to the DNA tetrahedron to obtain the normalized stability levels.

## DNA ligation experiments

Short duplexes with 20 and 30 bp with 3 or 4 nucleotide overhang were prepared by mixing 50 μM oligos and annealing them using temperature ramp from 90 °C to 20 °C with a temperature gradient of 1 °C/min in 1X PBS buffer (see Table S1). To measure the kinetics, 20 and 30 bp duplexes were mixed in equimolar ratio (0.5 μM) in buffer with final concentration of 1X T4 DNA ligase buffer (NEB), 1X BSA, 1 mM ATP. 1 μl T4 DNA ligase (40 units/μl) was added to the mixture. The reaction was terminated at the required time point by heat inactivation at 70 °C for 20 min. Then the reaction mixtures were mixed with the gel loading buffer (final concentration 1X) and were run in a 10% non-denaturing PAGE (29:1 acrylamide/bisacrylamide) at room temperature (150 V, 1 h). After electrophoresis, the gels were stained with GelRed (Biotium) and imaged using Bio-Rad Gel Doc XR+. Gel bands were quantified using ImageLabs software. The ligated product was quantified and normalized against the 50 bp marker band in the 10 bp-DNA ladder (Thermofisher, Catalog number: SM1313).

## Molecular dynamics simulations

Two end-to-end stacked 3-mer duplexes were pulled apart using molecular dynamics (MD) simulations to mimic CFM experiments. We evaluated the stacking parameters of two nucleic acid force fields: Amber-99 with Chen-Garcia correction[52] and parmbsc1[54]. The initial structures were constructed using the Molecular Operating Environment software[59] and consist of sequences 5'-CGX|xTC-3' and 3'-GCY|yAG-5', where | indicates the boundary of each duplex; Xx-Yy are either AA-TT, or AT-TA, respectively. The simulation system consisted of the two 3-mer DNA duplexes in a solution of ~66,000 water molecules and 177 $Na^+$ ions and 169 $Cl^-$ ions, enclosed in a 10 nm × 20 nm × 10 nm 3D periodic box. Water molecules were represented with the TIP4P model[60] and LINCS was used to restrain hydrogens bonded to heavy atoms[61]. Long-ranged electrostatic interactions were calculated using particle mesh Ewald (PME) algorithm[62]. The system was subjected to steepest-descent energy minimization, followed by NVT and NPT equilibration runs, 1 ns each maintained at 294 K using the velocity rescaling thermostat[63] and at 1 atm using the Berendsen barostat[64]. After equilibration, a MD pulling simulation was performed using a pulling force constant of 3200 kcal·$mol^{-1}$·$nm^{-2}$ to generate sixteen initial structures of the system with varying initial distances (0.475 nm & 0.5-1.2 nm in 0.05 nm intervals) between the pull groups (C1' atoms on one of the stacked base pair). Each of the 16 structures was then simulated for 2.6-2.8 ns/distance, totaling ~42 ns of production run per construct per forcefield. The simulations incorporated leap-frog algorithm with a 2 fs time-step in the NVT ensemble at 310 K using the velocity rescaling thermostat[63]. The base-pairing between the middle two base pairs and the ones closest to unstacked the 3' and 5' ends were maintained using distance restraints on the hydrogen bonds, with a force constant of 1195 kcal·$mol^{-1}$·$nm^{-2}$. System coordinates were stored every 2 ps. All MD simulations were performed using Gromacs-2020.01 package. The reversible work required to separate the 3-mer duplexes along the axis of the ds DNA is measured piecewise, at equilibrium, by calculating the potential mean force (PMF) across the distance intervals, extracted using the Weighted Histogram Analysis Method (WHAM) module[65].

## Reporting summary

Further information on research design is available in the Nature Portfolio Reporting Summary linked to this article.

## Data availability

Raw data used to generate graphs in this paper are included as supplemental information and in the source data file. Full gel images are included as supplemental figures. Original movie files will be shared upon request, but cannot easily be publicly available due to the large data size (~1 TB). Requests for non-commercial use of movie files can be made to the corresponding author and will be filled as soon as possible within 1 month. Source data are provided with this paper.

## Code availability

Matlab code used to determine dissociation times is provided as supplemental file along with sample data and instructions. The code is described in the "Methods" section and in ref. [32].

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

## Acknowledgements

The authors thank Andreas Karl from Thermo Fisher Scientific for providing the centrifuge main board allowing computer control. We thank Pan T.X. Li and Lifeng Zhou for providing feedback on the project. Research reported in this publication was supported by the National Institutes of Health through the National Institute of General Medical Sciences under award R35GM124720 to K.H. and R35GM133469 to A.C., and by the National Science Foundation under award MCB1651877 to A.C.

## Author contributions

The project was conceived and planned by K.H. and J.A.P., and super-vised by K.H. Single-molecule experiments were planned by J.A.P. and were carried out by J.A.P., K.T., A.H., and T.B. Single-molecule data analysis was performed by J.A.P. and K.T. DNA tetrahedra experiments were planned and carried out by A.R.C. DNA ligation experiments were planned by J.A.P. and carried out by J.A.P. and J.V. Molecular dynamics simulations and analysis were planned and carried out by K.K., S.V., and A.C. Technical support and instrument repairs were provided by A.H. The paper was written by J.A.P., A.R.C., and K.H. with MD content by K.K., S.V., and A.C., and with general input and editing from other authors.

## Competing interests

The corresponding author (K.H.) is an inventor on patents of the CFM instrument and use and has received licensing royalties related to those patents. The remaining authors declare no competing interests.
