## [Peer Review File · Nature Communications]

REVIEWER COMMENTS

Reviewer #1 (Remarks to the Author):

The authors characterize individual base-stacking energies in nucleic acids using the centrifuge force microscope (CFM), a custom-built instrument that is uniquely suited to quantify the strengths of single molecular interactions with exceptionally high throughput. They present a wealth of new information about the contributions of individual bases to the stacking energy, demonstrating convincingly not only the superiority of their methodology but also the importance of their results in different applications. The study looks thorough and solid, the presentation is concise, clear, and well-illustrated. I enjoyed reading this manuscript, and I believe that this work is of interest to a broad audience, in particular in various disciplines of nucleic acid science and engineering. The experimental approach and results should be interesting and accessible even to researchers who are not specialists in this field, and some of the reported values are likely to stand the test of time, perhaps even end up in textbooks.

I only have a few minor comments and questions:

- Perhaps I have overlooked this information, but I am wondering about the time resolution of the CFM measurements. How fast can successive images be acquired during an experimental run?

- In connection with the above question, different graphs of the raw data show counts of tethered molecules that were taken at different time intervals, even when the overall duration of experiments was similar. Is there a particular reason for this?

- How was time zero defined? Is this the time when the centrifuge reached the target RPM? How long did it take to spin up the centrifuge compared to the duration of (short) experiments? I don't believe that the exact starting time of each tether count matters for this measurement (correct?), but it would be useful to get a general idea.

- The authors state that multiple changes of the pixel-intensity variance of the image of a given bead were interpreted as successive failures of multiple tethers, and that such data were discarded, which is sensible practice. I am still wondering about the possibility that successive failures (especially right before the bead disappeared) might have been missed in cases where the delay between them was shorter than the time interval between video snapshots. Could the potential inclusion of such multiple binding events in the results be of concern? Some of the exponential decay data in the Supplementary Information look a little less convincing than most others; I am wondering if this might be due to the inclusion of a few multiple unbinding events?

- In connection with the above question, could one identify single-molecule binding based on the bead image alone? Two tethers will result in a significantly shorter distance of the bead from the glass coverslip than a single tether, which might be detectable due to the difference in the bead image, right?

- I think it is not unusual that the off-rate constants extracted from single-molecule force measurements based on the Bell-Evans model differ somewhat from the off-rate constants obtained in equilibrium situations. One might assume that this difference becomes smaller as the clamping force decreases (as long as the Bell-Evans model provides a good fit), but wouldn't even small forces select a "reaction coordinate" that not necessarily coincides with the force-free unbinding path? It would be good to briefly discuss this possibility.

- In the second sentence of the Discussion, should it be "interactions" instead of "iterations"?

Reviewer #2 (Remarks to the Author):

In this manuscript, Halvorsen and colleagues present measurements of the unbinding of DNA duplexes held under tension in a centrifuge force microscope (CFM), which they used to deduce the free energy for stacking of individual bases. By repeating for different bases, they found stacking energies for all 10 distinct combinations. They also looked at the effect of various chemical modifications to the nucleotides. They explored whether they could use differences in stacking energies to tune the stability of a DNA nanostructure and the kinetics of a ligation reaction. Finally, they looked at the implications of their measurements for evaluating the effectiveness of molecular dynamics force fields.

Accurate measurements of nucleic acid base-stacking energies are useful for a wide range of scientific and technological applications, hence the topic of this work is of suitable general interest. The approach the authors have taken is also very appealing, in that it seems relatively straightforward to implement experimentally and could be extended to explore a variety of related questions, such as characterizing the effects of the various conditions that are known to alter nucleic acid duplex stability (e.g. temperature, salt concentration, pH, ...). However, there are a number of concerns that need to be addressed in revision, related to the interpretation of the data and the significance of the results in the context of previous work.

Specific points to address:

1. The authors have framed their study as the first to measure the contribution to duplex stability from stacking between individual bases, emphasizing that most previous work looked at stacking between pairs of bases. However, this claim is not strictly true, nor is it clear why measurements of stacking in individual bases is a major advance over measurements of stacking in base-pairs.

With regards to the first point: the Croquette group used a magnetic tweezers assay to deduce the A|G stacking energy using a very similar experimental design. Granted, they measured only one stacking energy, but the extension from 1 to all 10 is quite obvious and would seem to be a clear case of an incremental (rather than substantial) advance. The authors neglect to cite this work in the introduction when discussing previous studies of stacking, mentioning it only briefly in the Discussion at the end of the manuscript, which does not seem appropriate for placing their work in proper context. The authors should provide a more complete and accurate description of prior work in the Introduction section, making the context of their own work clearer.

With regards to the second point: stacking stabilities for various base-pairing combinations have been measured previously. Given that stacking energies appear to be additive (see, for example, Ref. 22), could one not simply use the stacking energies for different combinations of base-pairs to deduce the energies for single-base stacks? If so, then it would seem to me that the only advantage of being able to measure single-base stacks is that one could directly test the assumption of additivity. The authors need to make a clearer and more compelling case for why their work represents a major advance over the previous work using base-pairs, since it is not obvious that the distinction the authors make between their measurements and those on base-pairs is meaningful. As part of responding to this point, the authors should look into the possibility of using base-pair data to deduce single-base stack energies, compare to their results, and test if they find the energies are indeed additive.

There are other aspects of the work that have precedent in the literature, such as the fact that the stacking energies are force-independent (discussed in Ref. 22). Please acknowledge these points.

2. Deducing the stacking energy: The authors claim that they are measuring the stacking energies directly through their approach, but that is not strictly correct. Instead, they are measuring rupture rates, using them to deduce the change in the barrier height, and then assuming that the transition state is native-like, so that the change in the energy of the duplex is the same as the change in the energy of the transition state. However, it is not at all obvious that this assumption is correct! In fact, given that shearing of a duplex by mechanical force involves significant distortion of the bond angles for the outermost bases (see for example Lang et al. Nature Methods 2004 and theoretical work by David Nelson and Pierre de Gennes), one might expect a distinct difference between the change in the energy of the duplex when changing the stacking vs the change in the energy of the transition state. Note that the problem arises here because the method the authors use does not allow for re-annealing of the

duplex, such that the on/off equilibrium cannot be observed (the same as in ref. 54 but unlike, for example, in ref. 22). This seems to me a fundamental problem with the approach that the authors need to address. It might explain, for example, why the stacking energies deduced in this work are systematically higher than the values obtained previously by methods that are able to observe on/off equilibria. Note that the results for base-pair stacking energies from force spectroscopy and gels did not show such an obvious systematic difference between each other.

3. The comparison to previous work in the Discussion section is perfunctory and should be considerably expanded with more quantitative detail. One ought to be able to add together the energies for stacking two bases to generate base-pair stacking energies. How well does that work when comparing to previous work from force spectroscopy and gels? If the estimates here are noticeably higher (as appears to be the case), there might very well be systematic errors in play (e.g. the issue mentioned in point 2 above), which should be discussed (and ideally evaluated).

4. Data presentation and fitting (including SI figures):

(i) For all graphs showing exponential decays, please plot most of them on a log scale, it will make the presentation and interpretation clearer. For any linear plots, please also add in a horizontal line at 0, so that the reader can see the decay to 0.

(ii) The fitting function includes a vertical offset term. There is no obvious physical justification such an offset should exist, since the experimental observable is counts of molecules—once all the tethers are broken, there should be identically 0 left. Please provide a physical justification for the use of an offset, or else remove it.

(iii) Some of the curves do not look completely exponential (there appear to be systematic deviations). A particularly obvious example is shown in Fig. 2g, but there are several others. It would be a good idea to perform some comparative statistical test to show that the single exponential fits better than other possibilities (e.g. double exponential, stretched exponential).

5. Precision of language: in a few places, the authors make statements that are not strictly true and should be corrected.

(i) Abstract line 1: nucleic acid structures are not just stabilized by base pairing and stacking interactions, there are also interactions with ions and ion clouds, as well as tertiary contacts (in RNA).

(ii) Abstract, results, discussion: regarding the measurements on “modified bases”, most of the modifications are not to the base but elsewhere on the nucleotide, please be more consistent in referring to modified nucleotides rather than bases.

(iii) Introduction, first paragraph: the title and abstract refer generally to nucleic acids, but the introduction starts off specifically with DNA, then brings RNA into the picture without ever acknowledging the important differences between them. Given that this manuscript does not, in fact,

study RNA stacking energies (see comment above), it would be better to change the title and abstract to discuss DNA only.

(iv) Introduction, first paragraph: the claim that base stacking interactions are underappreciated is somewhat overstated, given that it's been appreciated for decades that stacking interactions contribute very significantly to the nearest-neighbor duplex energies.

(v) Introduction, first paragraph: the implication that a high unfolding force seen for an RNA kissing-loop complex reflects a high thermodynamic stability is overly simplistic: strictly speaking, the unfolding force reflects the height of the energy barrier. In the case of the kissing loop experiment, the high mechanical stability arose mostly from the fact that the barrier was quite insensitive to force (low distance to the transition state), i.e. pulling on the stems was not very effective at breaking the loop interaction.

(vi) Introduction, first paragraph: for most of the examples given, stacking and base-pairing are *both* important. The intent of this paragraph seems to be to make the case that stacking energies have been sorely neglected in the field, but that seems to me an overstatement.

(vii) Results, line 165: the authors claim that their approach to deducing stacking free energies is "validated." Validating a new approach requires showing that it yields the same result as found using some other, established approach(es). The authors have not done so here, hence the claim should be re-phrased.

6. The measurements using a ribose on the stacked base are interesting, but not because they provide any indication of the stacking energy in RNA—they do not! They're interesting because they suggest that the higher stability of RNA is likely due entirely to higher stacking energies for A-form helices than B-form helices. Here, because only a single sugar moiety is changed to ribose, the helix is presumably still B-form. A more direct test of this picture would be to repeat the experiment with the construct converted into RNA.

7. For the application to DNA tetrahedral, the significance of this work is somewhat unclear. Are these structures normally made with gaps in the duplex? Why can't one just use the standard nearest-neighbor energies for ensuring the stability of the design? Is the idea that one can now adjust the stability of the structure via individual stacking energies if desired? What is the advantage taking such an approach, instead of adjusting the stability of the structure by tuning the sequence to alter the nearest-neighbor energies?

8. For the application to ligation reactions, again, it's unclear what the major advance is. Can't one already make predictions for the strength of the sticky-end interactions simply using tables of nearest neighbor energies?

9. For the comparisons with MD simulations, please clarify how the PMF is obtained and what assumptions are made in the analysis. The simulations are very short (ns scale), hence the timescales are

$\sim 10^9$ times faster than the experiments. Is the analysis using WHAM recovering an equilibrium free energy from the non-equilibrium simulation? That presumably introduces considerable systematic errors, which will complicate comparisons.

10. Measurements/analysis of unbinding rates: do the measurements and analysis take into account the time required to spin up the CFM to the final speed? Or are ruptures only counted once the final speed is reached? One assumes the latter, because otherwise the analysis is much more complex (given that the force-dependent survival time would need to be included during the ramp up). Please clarify these details of the measurements and analysis.

11. Figures:

Figure 1: The cartoons could be clearer. Perhaps the authors could illustrate the physical process of breaking the duplex by force, so that it's distinct from the thermal melting in panel c? Also, the justification for the energy diagram, in panel c is unclear: what evidence is there that the transition state is the same with/without stacking? The authors seem just to be assuming that this is true, but a key feature of the equilibrium measurements is that it doesn't matter what happens to the transition state.

Figure 3: in panel c, please report the slopes (with errors), to confirm that they are the same.

Figure 4c: please specify the source of the errors (presumably fitting error).

Figure S5: please clarify that the results at 0 pN are not measured, but extrapolated.

Reviewer #3 (Remarks to the Author):

The authors present a novel method with CFM to measure the individual stacking energies between two adjacent bases. The experimental design is clear and interesting. They provide stacking energies value within 0.2kcal/mol uncertainty for all base stacking orders and various modifications. The results could be helpful in many applications including DNA nanostructure, enzymatic ligation and MDs mentioned in the manuscript. However there are also some major points to refine.

1) The convergence must be evaluated before the calculation of base stacking energy.

2) As Amber-99 Chen-Garcia is optimized for RNA, how did its parameter be transferred for simulating d(T), more details need to be provided.

3) The temperature of MDs(310K) is not consistent with that of experiment(294K). At the same time, the concentration of ion is not same to that of experiment. This may induce undesirable inaccuracy and a bit strange that why not to keep the same temperature. In general, we should use the same condition of experiment and compare with the results.

** See Nature Portfolio's author and referees' website at www.nature.com/authors for information about policies, services and author benefits.

Response to Reviewers

We express our sincerest thanks to the editor and referees who handled this manuscript and provided thoughtful and constructive feedback. With this response to reviewers, we address the concerns in a point-by-point manner and present an improved manuscript based on the suggestions. We also rewrote our abstract to meet the style and length requirements of *Nature Communications* since the original manuscript was transferred from *Nature* and contained their unique abstract style. Our responses are in blue text with reviewer comments in black. Please note that reference numbering has changed from the original text.

The new abstract is pasted here:

“Base stacking interactions between adjacent bases in DNA and RNA are important for many biological processes and in biotechnology applications including drug development. Previous work has estimated stacking energies between pairs of bases, but contributions of individual bases has remained unknown. Here, we use a Centrifuge Force Microscope for high-throughput single molecule experiments to measure stacking energies between adjacent bases. We found stacking energies strongest between purines (G|A at -2.3 ± 0.2 kcal/mol) and weakest between pyrimidines (C|T at -0.5 ± 0.1 kcal/mol). Hybrid stacking with phosphorylated, methylated, and RNA nucleotides had no measurable effect, but a fluorophore modification reduced stacking energy. We experimentally show that base stacking can influence stability of a DNA nanostructure, modulate kinetics of enzymatic ligation, and assess accuracy of force fields in molecular dynamics simulations. Our results provide insights into fundamental DNA interactions that are critical in biology and can inform design in biotechnology applications.”

Reviewer #1 (Remarks to the Author):

The authors characterize individual base-stacking energies in nucleic acids using the centrifuge force microscope (CFM), a custom-built instrument that is uniquely suited to quantify the strengths of single molecular interactions with exceptionally high throughput. They present a wealth of new information about the contributions of individual bases to the stacking energy, demonstrating convincingly not only the superiority of their methodology but also the importance of their results in different applications. The study looks thorough and solid, the presentation is concise, clear, and well-illustrated. I enjoyed reading this manuscript, and I believe that this work is of interest to a broad audience, in particular in various disciplines of nucleic acid science and engineering. The experimental approach and results should be interesting and accessible even to researchers who are not specialists in this field, and some of the reported values are likely to stand the test of time, perhaps even end up in textbooks.

We thank the reviewer for the kind remarks! We hope the readers will also share the same enthusiasm once the paper is published.

I only have a few minor comments and questions:

- Perhaps I have overlooked this information, but I am wondering about the time resolution of the CFM measurements. How fast can successive images be acquired during an experimental run?

This is a fair point that we neglected to address clearly in the first version. The fastest rate of transfer for full-size images captured on our 5 MP camera is 2 frames per second (fps). The frame rate can be increased by reducing the size of image area. For this study, we captured image at 1 fps and going at a slower pace was possible due to the time scale of the experiments ranging from several minutes for weaker interactions to ~2hrs for stronger interactions. We have clarified this point with the following text:

Main text (Results paragraph 2):

“During a typical experiment, we observe tens to hundreds of tethered microspheres in a full field of view at 40x magnification at a rate of 1 frame per second (Fig. 2d).”

Methods (Constant force experiment protocol):

“For our 5MP camera the fastest transfer rate was 2 fps, and for these experiments we used 1fps and saved every fifth frame to reduce data size. This rate was appropriate for our experiments which ran for several minutes to ~2 hours.”

- In connection with the above question, different graphs of the raw data show counts of tethered molecules that were taken at different time intervals, even when the overall duration of experiments was similar. Is there a particular reason for this?

While data was collected at 1 second intervals for all the experiments, not all the experiments were run for the same amount of time. For example, for the constructs with A|G stack, we ran the centrifuge for ~2hrs until beads stop dissociating, while for weakest stack, C|T, this was achieved in less than half an hour and even shorter for the constructs without base-stack. When histograms were made for the dissociation times, we tried to keep similar number of bins for every experiment for consistent fitting of decay curve which makes the decay curve look like experiments were performed at different time intervals. We have some explanation to clarify this in the Methods – Data Analysis section.

“Raw data of dissociation times were plotted as histograms with bin widths chosen to maintain approximately the same number of bins for each data set even with overall time spans varying by more than one order of magnitude.”

- How was time zero defined? Is this the time when the centrifuge reached the target RPM? How long did it take to spin up the centrifuge compared to the duration of (short) experiments? I don't believe that the exact starting time of each tether count matters for this measurement (correct?), but it would be useful to get a general idea.

You are right on all accounts. Time zero was the time of the first image taken after the centrifuge reached the target RPM. For the fastest RPM, it took ~6 sec to reach the target speed from zero. In contrast, the shortest experiment lasted several minutes. As you suggested, the start time shouldn't matter once constant force is attained since it should follow first order kinetics. The exponential decay would look the same at any point after time zero as long as there is enough data.

Since you have brought up this point we think it is useful to mention in the methods as a clarification:

“For subsequent analysis, time zero was defined as the first frame where the final RPM was reached.”

- The authors state that multiple changes of the pixel-intensity variance of the image of a given bead were interpreted as successive failures of multiple tethers, and that such data were discarded, which is sensible practice. I am still wondering about the possibility that successive failures (especially right before the bead disappeared) might have been missed in cases where the delay between them was shorter than the time interval between video snapshots. Could the potential inclusion of such multiple binding events in the results be of concern? Some of the exponential decay data in the Supplementary Information look a little less convincing than most others; I am wondering if this might be due to the inclusion of a few multiple unbinding events?

-In connection with the above question, could one identify single-molecule binding based on the bead image alone? Two tethers will result in a significantly shorter distance of the bead from the glass coverslip than a single tether, which might be detectable due to the difference in the bead image, right?

This are fair points that we can address together. We have taken several reasonable measures to identify and exclude multiple tethered beads (including your suggested method), so we are confident that the vast majority of tethers we observe dissociating are single tethers.

1. We optimized the construct to bead ratio such that most beads don't form tethers. This will already bias the tethers to primarily be single tethers based on Poisson statistics.
2. All the tethered beads in the analysis were checked visually with the help of the matlab code provided, for the contrast of the beads and bead cross-section profile. The multiple tethered beads looks visually different due to shorter distance from the cover-slip. These beads were excluded from analysis.
3. Beads showing multiple shifts in image variance after analysis were excluded. Nearly all of the beads that looked different and were excluded in step 2, had observable two step breaking. We haven't observed that off focused beads breaking off as a single-event, which indicates that possibility of breaking two tethers within a second is very low, especially when force is applied in different geometry.

With these three measures combined, we are confident that multiply tethered beads are almost completely avoided from the analysis. Since this question came up, we will add some text to the methods to better explain the process (new text underlined).

"The identified beads were visually checked to ensure beads appeared to be single tethers. Rare anomalous non-spherical beads, closely clustered beads, and dirt or other objects wrongly identified as beads were excluded. Additionally, beads out of the typical focus were excluded due to the possibility of multiple tethers. Once beads were identified from an image at the start of the experiment, the software calculated the variance of the image intensity at the bead location for all the frames. When beads dissociate, it is indicated by the sharp drop in variance (i.e. high contrast to low contrast). Multiple drops in variance were rarely observed among analyzed beads due to our pre-screening, and were excluded from analysis due to the possibility of multiple tethers."

Still, this is experimental data, and it is unrealistic to think our data is perfect. We cannot exclude the possibility that rare anomalous data slipped through, but we have taken every reasonable effort to ensure our data is of the highest quality we can attain. Those in the single-molecule field know that anomalies do happen, and could be a result of rare multiple tethers, differently formed tethers, or unexpected break points. These may account for some of the (minor) imperfections you see in our data, especially in cases where some beads remain stuck at the end of the experiment, causing the decay to not end in exactly zero (as was also brought up by reviewer 2).

In response to your comment about the variation in fitting quality in some of the supplemental curves (echoed in part by reviewer 2), we believe this is attributable to the number of data in each set, as well as some anomalous long lived tethers as you propose. The less perfect fits are almost all from data sets with fewer data. Still, it is worth pointing out that the fits as a whole are objectively excellent – among the 36 data sets plotted in Figures S7 through S9, the mean R^2 value is 0.992 and the lowest among them is 0.974. Even ideal data in this case would have some statistical variation. We simulated an "ideal" experiment using an exponential distribution (using MATLAB `expnd` function) and then plotted and fit the resulting data. For sets with 50 data points (similar to our worst fitting datasets), the R^2 was typically between 0.98 and 0.99. For sets with 100 data points, the R^2 was >0.99 . We are plotting our data in a transparent way and feel that overall it is high quality.

Considering your comments and those of reviewer 2, we have added some appropriate text in the results where we first describe fitting the data:

“The data were well described by single exponential decays to determine off-rates at different forces, with R^2 values typically exceeding 0.99 (Fig. S2-S4). Some of the smaller data sets ($n < 100$) had R^2 values closer to 0.95. We note that the y -offset was allowed to be greater than zero, since some data had a small percentage of stuck beads which can arise from anomalous tethers.”

- I think it is not unusual that the off-rate constants extracted from single-molecule force measurements based on the Bell-Evans model differ somewhat from the off-rate constants obtained in equilibrium situations. One might assume that this difference becomes smaller as the clamping force decreases (as long as the Bell-Evans model provides a good fit), but wouldn't even small forces select a “reaction coordinate” that not necessarily coincides with the force-free unbinding path? It would be good to briefly discuss this possibility.

This is a good point. In combination with reviewer 2 a few issues with our model were brought up, so we added a paragraph in the discussion to address the assumptions and uncertainties:

“Given the comparison with previous results, it is worth exploring assumptions and uncertainties in our study. The work is largely predicated on the model in Fig. 1c, which assumes a 1D energy landscape with a transition state that doesn't change appreciably between constructs. In pulling experiments, the reaction coordinate is constrained to the 1D pulling direction, and a misalignment with the equilibrium reaction coordinate could bias the extrapolation to the zero-force off-rate. Also potentially biasing this extrapolation would be changes in the transition state, which could arise from physical differences such as mechanical forces distorting bond angles [56]. However, these concerns are largely alleviated by 1) our construct design, 2) evidence from past literature, and 3) our own results. We designed constructs to minimize differences; base stacked constructs are structurally identical (only differing by identity of a single base) and control constructs are structurally similar with the insertion of a small ssDNA gap. With this design, we expect extrapolation biases to equally affect all of the base stacked constructs and likely also the non-stacking controls, preserving the stacking energy estimates. Previous literature largely supports this model as well. Pulling on duplexes has been suggested to extrapolate to the equilibrium off-rate [57], even with variations in construct design, suggesting that the 1D model is a reasonable approximation. Further, the transition state only weakly depends on duplex length [35], and helicity and base pair tilt have been shown to not play a major role in duplex shearing [56], together suggesting that small structural variations do not alter the transition state appreciably. Our own experimental data in Figure 3 supports empirically this by exhibiting parallel slopes in the force dependence of different constructs. The close agreement of our non-equilibrium data with equilibrium measurements under force by Rieu et. al [55] also suggests our approach is valid.”

- In the second sentence of the Discussion, should it be “interactions” instead of “iterations”?

Good catch, thank you – we have corrected the word.

Reviewer #2 (Remarks to the Author):

In this manuscript, Halvorsen and colleagues present measurements of the unbinding of DNA duplexes held under tension in a centrifuge force microscope (CFM), which they used to deduce

the free energy for stacking of individual bases. By repeating for different bases, they found stacking energies for all 10 distinct combinations. They also looked at the effect of various chemical modifications to the nucleotides. They explored whether they could use differences in stacking energies to tune the stability of a DNA nanostructure and the kinetics of a ligation reaction. Finally, they looked at the implications of their measurements for evaluating the effectiveness of molecular dynamics force fields.

Accurate measurements of nucleic acid base-stacking energies are useful for a wide range of scientific and technological applications, hence the topic of this work is of suitable general interest. The approach the authors have taken is also very appealing, in that it seems relatively straightforward to implement experimentally and could be extended to explore a variety of related questions, such as characterizing the effects of the various conditions that are known to alter nucleic acid duplex stability (e.g. temperature, salt concentration, pH, ...). However, there are a number of concerns that need to be addressed in revision, related to the interpretation of the data and the significance of the results in the context of previous work.

We thank you for the interest in the manuscript, and your encouraging comments about the methodology and broad interest. Your concerns are well thought out (and got us thinking too!) and we have taken our best reasonable effort to address them below:

Specific points to address:

1. The authors have framed their study as the first to measure the contribution to duplex stability from stacking between individual bases, emphasizing that most previous work looked at stacking between pairs of bases. However, this claim is not strictly true, nor is it clear why measurements of stacking in individual bases is a major advance over measurements of stacking in base-pairs.

With regards to the first point: the Croquette group used a magnetic tweezers assay to deduce the A|G stacking energy using a very similar experimental design. Granted, they measured only one stacking energy, but the extension from 1 to all 10 is quite obvious and would seem to be a clear case of an incremental (rather than substantial) advance. The authors neglect to cite this work in the introduction when discussing previous studies of stacking, mentioning it only briefly in the Discussion at the end of the manuscript, which does not seem appropriate for placing their work in proper context. The authors should provide a more complete and accurate description of prior work in the Introduction section, making the context of their own work clearer.

We are actually big fans of that particular work by Croquette group (in fact hoping to integrate their imaging method into our next CFM iteration) and were not intending to minimize that contribution. We excluded it from the introduction simply because it was not background material for this work. It was published recently (Feb 2021), at a point after we were largely finished with data collection for our paper. The first author of our work presented preliminary data from this project on A|G and A|C base stacking at *Biophysical Meeting* in Feb 2019 - two years before the Croquette paper came out (see published abstract below).

1365-Pes

A Single-Molecule Investigation on Interfacial Base-Stacking Interaction using a Centrifuge Force Microscope
Jibin Abraham Punnoose.

The RNA Institute, SUNY at Albany, Albany, NY, USA.

Single-molecule force experiments can be critical in providing a mechanistic understanding of biomolecular interactions. The recently developed Centrifuge Force Microscope (CFM) enables massively parallel single-molecule force manipulation with a low cost and easy to use instrument. Experiments are performed by subjecting surface tethered microspheres to centrifugal force while observing their motion with a microscope objective coupled to a CMOS camera. In this project, we have developed a plug-and-play CFM module that is self-contained in a commercial centrifuge bucket, and is able to live-stream images to an external computer during centrifugation. The whole system including data acquisition is controlled wirelessly through a LabVIEW interface. Using this system, we have been investigating the influence of the interfacial DNA base stacks on the stability of DNA structures held together by sticky-end hybridization. Through systematic elimination of interfacial base-stacks on one or either strand we were able to analyze the contribution of DNA base-stacking. DNA duplexes held together only by base-pairing showed significantly faster force-dependent dissociation rates compared to those held together by the same number of base-pairs with interfacial base-stacks. The high-throughput CFM can collect hundreds of data points in a single minutes-long experiment, enabling us to probe many different duplex variants and conditions. Additionally, the simple operation of our instrument facilitates use by undergraduate researchers, introducing a new generation to biophysics.

So in the scheme of incremental advances, it seems that both groups worked on it concurrently and without knowledge of each other. Considering this, we still feel like it appropriately cited in the discussion.

With regards to the second point: stacking stabilities for various base-pairing combinations have been measured previously. Given that stacking energies appear to be additive (see, for example, Ref. 22), could one not simply use the stacking energies for different combinations of base-pairs to deduce the energies for single-base stacks? If so, then it would seem to me that the only advantage of being able to measure single-base stacks is that one could directly test the assumption of additivity. The authors need to make a clearer and more compelling case for why their work represents a major advance over the previous work using base-pairs, since it is not obvious that the distinction the authors make between their measurements and those on base-pairs is meaningful. As part of responding to this point, the authors should look into the possibility of using base-pair data to deduce single-base stack energies, compare to their results, and test if they find the energies are indeed additive.

This is a good point that I think we need to better explain this in the paper. There are a number of advantages in our method and our results: 1) Similar to the Croquette method, ours is relatively direct (meaning relatively few assumptions and minimal disturbance to natural DNA structure) when compared to previous work. In principle this should make our results more accurate, 2) while I agree the energies should be additive [ref. 22 fairly assumes them to be, but provides no evidence], measuring dyad pairs does not allow deconvolution of the individual contributions. This is because the stacking of pairs in previous experiments requires the pairs to follow base pairing rules, so not all combinations of pairs are accessible for experimentation. The only 2 pairs that could be deconvolved would be GC:CG and AT:TA because these would form two identical stacks, so one could presumably divide by 2 to get the individual stack. Other combinations such as CG:AT produce one value but there is no way to determine which of the two stacks is the dominant one, or what the relative contributions of each are. We certainly don't want to minimize the importance of past work, which is why we treaded lightly on this point, but to our knowledge nobody could have produced a rank-ordered list of base stacking energies as we have done in figure 4.

The importance of the individual values is best illustrated in Figure 6, because in these examples the individual base stacks can be manipulated. The base stacking dyad values do not apply well to these types of biotechnology applications for the deconvolution reasons just mentioned.

To more clearly explain the benefit of the measuring individual base stacks we have added some new text in the introduction:

“These studies have made immense contributions to our knowledge, but their designs and experimental constraints precluded the measurement of base stacking between two individual bases rather than pairs of bases. This has prevented knowing stacking energies between A and C for example, because in the context of a duplex this would be necessarily paired with stacking of the hybridized bases (T and G in this example). In these cases, a single energy value obscures the relative contributions of two or more interaction energies that cannot readily be deconvolved. Overall, the lack of data on individual base stacking interactions can limit informed design in biotechnology and synthetic biology where short engineered contacts are formed between various DNA or RNA strands.”

There are other aspects of the work that have precedent in the literature, such as the fact that the stacking energies are force-independent (discussed in Ref. 22). Please acknowledge these points.

We have added this point in the main text:

“Similar force-independence has been previously noted in the literature [22].”

2. Deducing the stacking energy: The authors claim that they are measuring the stacking energies directly through their approach, but that is not strictly correct. Instead, they are measuring rupture rates, using them to deduce the change in the barrier height, and then assuming that the transition state is native-like, so that the change in the energy of the duplex is the same as the change in the energy of the transition state. However, it is not at all obvious that this assumption is correct! In fact, given that shearing of a duplex by mechanical force involves significant distortion of the bond angles for the outermost bases (see for example Lang et al. Nature Methods 2004 and theoretical work by David Nelson and Pierre de Gennes), one might expect a distinct difference between the change in the energy of the duplex when changing the stacking vs the change in the energy of the transition state. Note that the problem arises here because the method the authors use does not allow for re-annealing of the duplex, such that the on/off equilibrium cannot be observed (the same as in ref. 54 but unlike, for example, in ref. 22). This seems to me a fundamental problem with the approach that the authors need to address. It might explain, for example, why the stacking energies deduced in this work are systematically higher than the values obtained previously by methods that are able to observe on/off equilibria. Note that the results for base-pair stacking energies from force spectroscopy and gels did not show such an obvious systematic difference between each other.

This is probably the hardest point to address to everyone’s satisfaction and was also brought up in a slightly different way by reviewer 1.

We concede that our description of “direct measurement” was not exactly accurate as you correctly point out, and we have revised that. Our intention was to convey the simplicity of the experimental methodology, especially in contrast with some of the work before us.

We contend that the major assumption is whether the sets of duplexes with base stacking and without base stacking have approximately the same reaction pathway and transition state. Phenomenologically, there are many evidences to suggest this is approximately true. The transition state in DNA shearing experiments has previously been found to be relatively insensitive to the length of the duplex being sheared [Strunz et. al., PNAS 1999]. In the Lang et. al. Nature Methods paper you refer to, they also found evidence that extrapolating to zero force gave near equilibrium results. Furthermore, the recent Croquette experiment which is closest in design to ours but does allow reannealing agrees with our measurements within error. We think this agreement suggests that our initial assumptions are largely valid.

In comparing with previous work, we again tread lightly to not diminish those efforts, which have been foundational and influential for our work. But going back to the “direct” comment I do believe that our methodology requires fewer assumptions to arrive at our results than the previous body of work to date. The work from Kamenetskii group requires assumptions about the effect of urea denaturant, the effect of the forces (or lack thereof) in gel electrophoresis, and the effect of bending energy on kinked vs. unkinked equilibrium. The work from Dietz group assumes perfect formation (and stability) of relatively complex DNA origami structures with sharp bending at the measured interfaces, assumes a completely passive linking tether, assumes perfect contact across several interfaces, and assumes multiple contacts share force evenly. It is worth pointing out that failure of most if not all of these assumptions would result in measuring a weaker base stacking interaction than the true value. We have also bought and used only PAGE purified oligos so that we can be sure the terminal base is present, which is not really addressed in previous literature. It has been unclear from previous studies what steps (if any) were taken to ensure purity of the base stacking interface.

To address your points in the text, we have added a new paragraph in the discussion that clearly states our assumptions and explains our reasoning.

“Given the comparison with previous results, it is worth exploring assumptions and uncertainties in our study. The work is largely predicated on the model in Fig. 1c, which assumes a 1D energy landscape with a transition state that doesn’t change appreciably between constructs. In pulling experiments, the reaction coordinate is constrained to the 1D pulling direction, and a misalignment with the equilibrium reaction coordinate could bias the extrapolation to the zero-force off-rate. Also potentially biasing this extrapolation would be changes in the transition state, which could arise from physical differences such as mechanical forces distorting bond angles [56]. However, these concerns are largely alleviated by 1) our construct design, 2) evidence from past literature, and 3) our own results. We designed constructs to minimize differences; base stacked constructs are structurally identical (only differing by identity of a single base) and control constructs are structurally similar with the insertion of a small ssDNA gap. With this design, we expect extrapolation biases to equally affect all of the base stacked constructs and likely also the non-stacking controls, preserving the stacking energy estimates. Previous literature largely supports this model as well. Pulling on duplexes has been suggested to extrapolate to the equilibrium off-rate [57], even with variations in construct design, suggesting that the 1D model is a reasonable approximation. Further, the transition state only weakly depends on duplex length [35], and helicity and base pair tilt have been shown to not play a major role in duplex shearing [56], together suggesting that small structural variations do not alter the transition state appreciably. Our own experimental data in Figure 3 supports empirically this by exhibiting parallel slopes in the force dependence of different constructs. The close agreement of our non-equilibrium data with equilibrium measurements under force by Rieu et. al [55] also suggests our approach is valid.”

3. The comparison to previous work in the Discussion section is perfunctory and should be considerably expanded with more quantitative detail. One ought to be able to add together the energies for stacking two bases to generate base-pair stacking energies. How well does that work when comparing to previous work from force spectroscopy and gels? If the estimates here are noticeably higher (as appears to be the case), there might very well be systematic errors in play (e.g. the issue mentioned in point 2 above), which should be discussed (and ideally evaluated).

We feel that this has now been addressed in the previous point. We are confident that our results are more accurate than those of our predecessors for reasons we mentioned in response to your previous point. We don’t think it is constructive for us to explain (in the main text) all of the reasons

why we think the previous studies have underestimated the base stacking measurements or to point out flaws. It's also clear from your thoughts that not everyone might agree with us. I hope you will find that we've clearly made our case and will let readers make their own judgement.

4. Data presentation and fitting (including SI figures):

(i) For all graphs showing exponential decays, please plot most of them on a log scale, it will make the presentation and interpretation clearer. For any linear plots, please also add in a horizontal line at 0, so that the reader can see the decay to 0.

This is a matter of opinion which has also been the subject of countless internal arguments amongst the authors. We have landed on the linear as the clearest representation for a general (non single-molecule) audience, even though the more physics-minded among us prefer the logarithmic plots. We have added the horizontal line to illustrate the "0"

(ii) The fitting function includes a vertical offset term. There is no obvious physical justification such an offset should exist, since the experimental observable is counts of molecules—once all the tethers are broken, there should be identically 0 left. Please provide a physical justification for the use of an offset, or else remove it.

This is a good point. The physical justification is one we didn't mention – that there are some anomalous beads which don't leave, even after a long time of pulling. This make up a very small percent, hence the typically 0-3% offset. We will add this to the text, and refine the fitting to ensure that it doesn't go negative, which certainly has no physical basis.

"The data were well described by single exponential decays to determine off-rates at different forces, with R^2 values typically exceeding 0.99 (Fig. S2-S4). Some of the smaller data sets ($n < 100$) had R^2 values closer to 0.95. We note that the y -offset was allowed to be greater than zero, since some data had a small percentage of stuck beads which can arise from anomalous tethers."

(iii) Some of the curves do not look completely exponential (there appear to be systematic deviations). A particularly obvious example is shown in Fig. 2g, but there are several others. It would be a good idea to perform some comparative statistical test to show that the single exponential fits better than other possibilities (e.g. double exponential, stretched exponential).

We have partly addressed this with our previous point, which was also mentioned by reviewer 1. Overall, the exponential provides a good fit and is the most reasonable physical model for the experiment. Adding more fitting parameters such as a modified exponential could fit better but then won't have a reasonable physical explanation. As we have stated with reviewer 1, the data is not perfect – it is experimental data. We think the simplest model along with a transparent description of our limitations is most reasonable.

I will point out that Figure 2g is meant to show the workflow and only contains 16 data points. It is a good illustration here of how small data sets can make the fitting look imperfect. We have clarified in that figure caption that the graph is based on 16 data points.

5. Precision of language: in a few places, the authors make statements that are not strictly true and should be corrected.

(i) Abstract line 1: nucleic acid structures are not just stabilized by base pairing and stacking interactions, there are also interactions with ions and ion clouds, as well as tertiary contacts (in RNA).

In the rewriting of the abstract to follow Nat. Comm guidelines this sentence was removed.

(ii) Abstract, results, discussion: regarding the measurements on “modified bases”, most of the modifications are not to the base but elsewhere on the nucleotide, please be more consistent in referring to modified nucleotides rather than bases.

Good catch, we have altered the language appropriately in the abstract and main text.

(iii) Introduction, first paragraph: the title and abstract refer generally to nucleic acids, but the introduction starts off specifically with DNA, then brings RNA into the picture without ever acknowledging the important differences between them. Given that this manuscript does not, in fact, study RNA stacking energies (see comment above), it would be better to change the title and abstract to discuss DNA only.

We do agree there was some awkward back and forth between DNA and RNA that we have now fixed. Regarding your suggestion to commit to “DNA” rather than “nucleic acids”, we have also argued internally about this. However, since the methodology applies generically to nucleic acids (as may some of the intro and discussion) we would prefer to keep the more general version. And while you do have a point with your RNA vs. DNA base stacking comment above, we still did test a non-DNA nucleotide so DNA doesn’t accurately encompass everything either.

(iv) Introduction, first paragraph: the claim that base stacking interactions are underappreciated is somewhat overstated, given that it’s been appreciated for decades that stacking interactions contribute very significantly to the nearest-neighbor duplex energies.

We concede this point. We have removed the offending text.

(v) Introduction, first paragraph: the implication that a high unfolding force seen for an RNA kissing-loop complex reflects a high thermodynamic stability is overly simplistic: strictly speaking, the unfolding force reflects the height of the energy barrier. In the case of the kissing loop experiment, the high mechanical stability arose mostly from the fact that the barrier was quite insensitive to force (low distance to the transition state), i.e. pulling on the stems was not very effective at breaking the loop interaction.

You are right on this point, but we also didn’t claim a high thermodynamic stability in our wording. Our sentence was: “An interesting example is a minimal RNA kissing complex, with only 2 canonical base pairs but unusually high mechanical stability (similar to a ~10 bp duplex) [1] attributed largely to base stacking interactions [2,3].” I have gone back and everything in that sentence seems supported by the title and abstract of the original paper [1]. I’m hesitant to get too much into biophysical details for the beginning of the introduction. I love this example because I too was shocked by these results, even coming from a biophysics background. It is actually the

example that first got me interested in studying base stacking in more detail, so I'm inclined to leave it.

(vi) Introduction, first paragraph: for most of the examples given, stacking and base-pairing are **both** important. The intent of this paragraph seems to be to make the case that stacking energies have been sorely neglected in the field, but that seems to me an overstatement.

Yes, we agree in hindsight that might have been overstated and we have tried to change the tone. We deleted the sentence fragment that *"base stacking interactions are sometimes overlooked"*

(vii) Results, line 165: the authors claim that their approach to deducing stacking free energies is "validated." Validating a new approach requires showing that it yields the same result as found using some other, established approach(es). The authors have not done so here, hence the claim should be re-phrased.

We deleted the word "validated" to be more accurate.

6. The measurements using a ribose on the stacked base are interesting, but not because they provide any indication of the stacking energy in RNA—they do not! They're interesting because they suggest that the higher stability of RNA is likely due entirely to higher stacking energies for A-form helices than B-form helices. Here, because only a single sugar moiety is changed to ribose, the helix is presumably still B-form. A more direct test of this picture would be to repeat the experiment with the construct converted into RNA.

This is a great point, and we have added this to the discussion. We have adjusted our language to be more precise about what we did and what it means. We definitely have our future sights set on the RNA-RNA base stacking but the cost and time to do that work is prohibitive to add on to this (already very large) paper.

"Our data seems suggestive that RNA base stacking may not be appreciably different from DNA, but with the caveat that our measurement inserted an RNA nucleotide into a DNA duplex, which is known to have a different structural form than RNA duplexes. Further work will be needed to clarify differences in RNA and DNA stacking, and the role of different chemical modifications on base stacking."

7. For the application to DNA tetrahedral, the significance of this work is somewhat unclear. Are these structures normally made with gaps in the duplex? Why can't one just use the standard nearest-neighbor energies for ensuring the stability of the design? Is the idea that one can now adjust the stability of the structure via individual stacking energies if desired? What is the advantage taking such an approach, instead of adjusting the stability of the structure by tuning the sequence to alter the nearest-neighbor energies?

You landed on the right idea here, terminal stacking energies can be used to adjust the stability of the structures. This cannot be pre-determined by nearest-neighbor because it is a combination of the 4 bp sticky end (whose energy can be estimated by nearest-neighbor) plus the two available terminal base stacking interactions that extend from the termini of the sticky ends to interface with the rest of the structure. The second part is exactly what was tested here and the part that cannot be predicted by nearest neighbor.

The point of the exercise was to 1) show that our results could be demonstrated in a real world example, which we did by showing that a G|A stack stabilizes a tetrahedron more than an A|T stack, and 2) to illustrate how base stacking provides a new element of design for DNA nanostructures. The point is that when you join things together with short overhangs (which is typical in DNA nanotech), that you can't (or shouldn't) ignore the effect of the terminal base stack. It makes a difference as we've clearly shown. We have added a sentence in the discussion to touch on this:

“The results would not be readily predictable from previous data or nearest neighbor approximations, and yet they provide support for both the magnitude and future utility this base stacking knowledge.”

8. For the application to ligation reactions, again, it's unclear what the major advance is. Can't one already make predictions for the strength of the sticky-end interactions simply using tables of nearest neighbor energies?

Similar answer to the previous. The nearest neighbor energies can estimate the stability of the base paired sticky end, but they cannot account for the terminal base stacking that lie outside the of the sticky end.

9. For the comparisons with MD simulations, please clarify how the PMF is obtained and what assumptions are made in the analysis. The simulations are very short (ns scale), hence the timescales are $\sim 10^9$ times faster than the experiments. Is the analysis using WHAM recovering an equilibrium free energy from the non-equilibrium simulation? That presumably introduces considerable systematic errors, which will complicate comparisons.

We share the reviewer's concern that equilibrium experiments should be compared to equilibrium simulations. As the reviewer points out, the experiments all involve shearing of an 8-bp duplex which occurs at a much slower timescale than the simulations. However, the effect of this duplex is then subtracted out to yield just the effective contribution from single-base stacking. Rather than simulating the slow shearing of the 8-bp duplex only to then subtract out its net contribution, we simulated just the unstacking of DNA duplexes using umbrella sampling. The reversible work required to separate two blunt-end dsDNA constructs along a linear reaction coordinate is measured piecewise, at equilibrium, and this is a ns timescale event at the forces measured and therefore readily converged in ns-timescale simulations. Therefore, the resulting PMFs provide a direct calculation of the free-energy barrier for dissociation conceptualized in figure 1C leading to the most direct comparison to the experimentally measured energies. We have added the following to the molecular dynamics methods section to clarify this point –

“The reversible work required to separate the 3-mer duplexes along the axis of the ds DNA is measured piecewise, at equilibrium, by calculating the potential mean force (PMF) across the distance intervals, extracted using the Weighted Histogram Analysis Method (WHAM) module [66].”

10. Measurements/analysis of unbinding rates: do the measurements and analysis take into account the time required to spin up the CFM to the final speed? Or are ruptures only counted once the final speed is reached? One assumes the latter, because otherwise the analysis is much more complex (given that the force-dependent survival time would need to be included during the ramp up). Please clarify these details of the measurements and analysis.

This point was also brought up by reviewer 1 and we apologize for the lack of clarity here. The analysis is all done at constant speed/force, and we have added to the text to better explain this. *“For subsequent analysis, time zero was defined as the first frame where the final RPM was reached.”*

11. Figures:

Figure 1: The cartoons could be clearer. Perhaps the authors could illustrate the physical process of breaking the duplex by force, so that it's distinct from the thermal melting in panel c? Also, the justification for the energy diagram, in panel c is unclear: what evidence is there that the transition state is the same with/without stacking? The authors seem just to be assuming that this is true, but a key feature of the equilibrium measurements is that it doesn't matter what happens to the transition state.

We can see this might lead to confusion. Unsure of how best to convey this, we've redrawn the cartoon in (d) to distinguish that the force unbinding and make it distinct from c as you suggest.

Regarding the justification of the model, you're right that it is an assumption. We assert that this simplest model has good justification both in previous literature and in our own data. In the literature, Strunz et. al. (PNAS, 1999) sheared different length duplexes and found only a weak dependence of the transition state on the number of bp (<1 Angstrom per bp). Our disruption is arguably less than what would be expected from a full additional bp, so the movement of the transition state would be negligible. To better convey this to the reader, we have made changes to the text to explicitly state the model and the assumptions (new text underlined).

“In the simplest model, this terminal base stack strengthens the interaction and lowers the energy of the bound state (Fig. 1c). The minimal disturbance of the interacting region in the duplex suggests that the transition state should not be appreciably disturbed, an assertion that is also supported by previous work that finds only a weak dependence of the transition state to the length of the duplex [35].”

Our own data also support the simplest model presented in Figure 1. We have added new text to better explain how our results support the model presented in 1c (new text underlined).

“We observed that force-dependent off rates were easily distinguishable between the constructs but followed identical slopes. Since the slopes are related to the position of the transition state, the consistent slopes support our model in Fig. 1c.”

It is a fair point that equilibrium measurements don't require this assumption, but also slightly off-topic because 1) many previous works were also not at equilibrium, or were at equilibrium at a condition that required extrapolation to normal experimental conditions (e.g. temperature, buffer, force); 2) no previous work has been able to measure these individual base stacks (excluding Croquette paper which was done concurrently); and 3) prior work required other assumptions to arrive at their results, some of which we think are less defensible than ours.

Figure 3: in panel c, please report the slopes (with errors), to confirm that they are the same.

We have now reported the slopes with errors in the caption.

Figure 4c: please specify the source of the errors (presumably fitting error).

The source of the errors is mostly in the standard deviation from the 3 data sets for each condition. We have made sure the errors are clearly described in the figure legend.

Figure S5: please clarify that the results at 0 pN are not measured, but extrapolated.

Good catch – we have amended the figure caption to explicitly say this:

“Figure S5: ΔG_{stack} determined for A|C and A|T base-stack at various forces. ΔG_{stack} at thermal equilibrium (zero force) was obtained from fitting Force - ΔG_{stack} data with Bell-Evans model (Figure 3c).”

Reviewer #3 (Remarks to the Author):

The authors present a novel method with CFM to measure the individual stacking energies between two adjacent bases. The experimental design is clear and interesting. They provide stacking energies value within 0.2kcal/mol uncertainty for all base stacking orders and various modifications. The results could be helpful in many applications including DNA nanostructure, enzymatic ligation and MDs mentioned in the manuscript. However there are also some major points to refine.

Thank you for your encouraging words. You have brought up a number of good points and we are happy to refine the MD aspects as you suggest.

1) The convergence must be evaluated before the calculation of base stacking energy.

The convergence of the simulations was indeed evaluated, and the calculated stacking energies do not show appreciable change in simulations 2-5 times longer than the one used for analysis. To illustrate this point, we have included a convergence plot in the SI [Figure S18].

2) As Amber-99 Chen-Garcia is optimized for RNA, how did its parameter be transferred for simulating d(T), more details need to be provided.

The reviewer makes an excellent point that DNA and RNA parameters are, in general, not readily interchangeable, as they prefer different helical geometries due to torsions and sugar puckers and have different base-pairing properties. However, in this specific case where single-based unstacking is the only interaction being probed, this is dictated solely by the VdW parameters of the nucleobases and the water model, which are identical for both DNA and RNA in the AMBER-derived force-fields. The charges of the d(T) base are taken from the AMBER-DNA forcefield using the VdW nucleobase-parameters of the Chen-Garcia RNA force-field which corrects for overstacking but was not parameterized in a base-specific manner. The dsDNA duplexes are restrained in their ideal geometries, effectively isolating the stacking interaction from any inaccuracies in the base-pairing or torsional preferences of the force-field.

3) The temperature of MDs(310K) is not consistent with that of experiment(294K). At the same time, the concentration of ion is not same to that of experiment. This may induce undesirable inaccuracy and a bit strange that why not to keep the same temperature. In general, we should use the same condition of experiment and compare with the results.

We completely agree that the simulation should match the experiment as much as feasible, and we have therefore re-run all simulations at 294K with the same excess ion concentration as the experiment, which now replaces the data presented in figure 6(n,p). The methods section has been updated as well. The new data does not alter any of the conclusions of the manuscript.

REVIEWER COMMENTS

Reviewer #1 (Remarks to the Author):

The authors have satisfactorily addressed my concerns in this revision. Congratulations on this nice work!

Reviewer #2 (Remarks to the Author):

The authors have addressed most of the concerns satisfactorily and the manuscript is definitely improved, but the responses on a few of the points were not adequate, in my view, and further revisions are warranted. They are relatively minor in terms of the work required, but important conceptually.

1. The most important concern relates to the assumption that the barrier is the same with and without stacking (model in Fig 1c). The issue here is that the assumption is required in order to back out the stacking free energies from the rate changes. The authors cite the fact that the slopes of the force-dependent rates were the same for the constructs with stacking and the control without, and "Since the slopes are related to the position of the transition state, the consistent slopes support our model in Fig. 1c." However, this claim is incorrect, or at least incomplete: the slopes only tell us the distance to the barrier, not the barrier height. In fact, the Bell-Evans model yield no information at all about barrier heights, since the barrier height is not a parameter in the model! If the transition state for base stacking/unstacking does have any sequence dependence (which is certainly a possibility), then the effective activation energy difference between the experiment and control would not be the value calculated from the equation in Fig. 1d. This seems like a very plausible source of systematic, sequence-dependent error on the final values for the stacking energies. The authors current treatment of this issue is not sufficiently nuanced, it should be discussed more carefully with appropriate caveats about how the assumptions being made might not hold.

On a related note, the authors did not report the Δx^\ddagger value they found from the Bell-Evans fits. How did they compare to the values found, say, for pair stacking in Ref. 22 or duplex shearing in Ref. 57? A small value for Δx^\ddagger would validate more quantitatively the assumption that there is not much distortion in the transition state.

2. Regarding the fact that some of the curves aren't all that well fit by single exponentials: I'm not convinced by the authors' claim that it's just an artifact of small datasets. Many cases do indeed seem to be just that (e.g. Fig. 2g, some of Fig. S4, some of Fig. S8). However, in other cases there is a systematic effect where the initial drop-off is faster than the single-exponential fit and the later drop-off is slower even when there are well over 100 molecules (e.g. in Fig. 3b A|T at 15 and 20 pN, in Fig. 4 C|C and C|T blue curves, in Fig. S4 all replicates at 15 pN, in Fig. S7 A|T replicate 2, in much of Fig. S8). Some of these cases have ~500 molecules. I appreciate that of course there will be noise, but one would expect noise to be less systematic! Despite the good R^2 values, I expect that applying something like the non-parametric Wald-Wolfowitz runs test would find that an appreciable number of these fits fail the test. The concern of course is that some systematic error is thereby introduced into the analysis of the stacking energies because the rates are not correct.

One possible explanation is that the slow tail represents multiple tethers. That would require a more sizable fraction of the tethers to be multi-tethers than one would expect, given what the authors did to try to minimize multiple tethers, but it could account for the inconsistencies in the appearance of the effect (e.g. only one of the replicates at 15 pN in Fig. S3, with other 500 molecules, shows the non-single-exponential decay). It might be quite difficult to determine the source of this effect, but the authors should at least acknowledge it as a possible bias in the discussion section.

3. I think the authors have explained more clearly why their work represents an important advance

over previous results. Nevertheless, the authors' insistence on omitting a mention of the work by Rieu et al. measuring base stacking energies in the introduction when outlining the previous efforts at investigating the problem the authors are addressing is not acceptable from the point of view of scholarship. The authors' contention that the paper by Rieu et al. was not published when the work was done and hence was not part of the background of the work is irrelevant: the authors are publishing their own work now, not two years ago (prior to publication of Rieu et al.), hence the paper by Rieu et al. is part of the existing scientific literature that must be taken into account when describing the context/background of what is known about base stacking and how it has been investigated. It would be very easy to include a mention of this reference in either paragraph 2 or possibly paragraph 3, pointing out that it introduced a promising method to measuring single-base stacking but did not apply it to systematically characterize stacking energies.

4. The authors say that they don't think it is constructive to explain the reasons why previous measurements have underestimated stacking energies. I disagree—in addition to discussing some of the features of the current measurement that might bias the results, it's valuable to mention some of the features that might bias the previous experiments. I doubt that those factors have been discussed much in the literature, since disagreement with other results did not exist when those papers were published. The discussion does not need to be extensive, but I think it will be a helpful contribution to understanding the significance of this paper.

5. Something I did not notice previously: the grey and blue curves in Fig. 4 do not seem to be defined in the figure caption, this should be corrected.

Response to Reviewer 2

Our responses are in blue text with reviewer comments in black. Note that reference numbers have changed.

Reviewer #2 (Remarks to the Author):

The authors have addressed most of the concerns satisfactorily and the manuscript is definitely improved, but the responses on a few of the points were not adequate, in my view, and further revisions are warranted. They are relatively minor in terms of the work required, but important conceptually.

We are happy that you agree the manuscript is improved, and here we work to address the remaining points. We appreciate the reasonableness of reviewer 2, even as we navigate some minor disagreements.

1. The most important concern relates to the assumption that the barrier is the same with and without stacking (model in Fig 1c). The issue here is that the assumption is required in order to back out the stacking free energies from the rate changes. The authors cite the fact that the slopes of the force-dependent rates were the same for the constructs with stacking and the control without, and “Since the slopes are related to the position of the transition state, the consistent slopes support our model in Fig. 1c.” However, this claim is incorrect, or at least incomplete: the slopes only tell us the distance to the barrier, not the barrier height. In fact, the Bell-Evans model yield no information at all about barrier heights, since the barrier height it not a parameter in the model! If the transition state for base stacking/unstacking does have any sequence dependence (which is certainly a possibility), then the effective activation energy difference between the experiment and control would not be the value calculated from the equation in Fig. 1d. This seems like a very plausible source of systematic, sequence-dependent error on the final values for the stacking energies. The authors current treatment of this issue is not sufficiently nuanced, it should be discussed more carefully with appropriate caveats about how the assumptions being made might not hold.

On a related note, the authors did not report the Δx^\ddagger value they found from the Bell-Evans fits. How did they compare to the values found, say, for pair stacking in Ref. 22 or duplex shearing in Ref. 57? A small value for Δx^\ddagger would validate more quantitatively the assumption that there is not much distortion in the transition state.

This is a fair point. We believe the vast majority of evidence points to the assumptions holding, but we understand they are still assumptions and that if they aren't true it would/could affect the outcome. We did neglect in our previous response to include one important point which was intended, namely that previous work has found that the on-rate is the same for stacked and unstacked duplexes (Rieu et. al.), and that rebinding rates are sequence independent (Kilchherr et. al.). This provides further evidence supporting model in figure 1c. As to the Δx^\ddagger values, we now added these values in the figure 3 caption, and they match within error to Lang et. al., ~ 0.5 nm $\pm \sim 10\%$. Those values found in Kilchherr et. al. were of similar magnitude, but with a relatively large spread from 0.29 to 2.37 nm.

To address these points in the text we have:

- added Δx^\ddagger values in the figure caption
- explicitly stated that deviations from our assumptions in Figure 1c would affect our end results

- added discussion of previous work showing insensitivities of on-rates to stacked configurations
- mentioned agreement of our transition state position relative to other work.

This is reflected in the text below (underlined):

“Also potentially biasing this extrapolation would be changes in the transition state, which could arise from sequence dependency of the transition state or physical differences such as mechanical forces distorting bond angles [56]. Any deviations from the idealized model in Figure 1c could cause systematic errors that would affect our end results of base stacking energies. However, we believe these concerns are largely alleviated by 1) our construct design, 2) evidence from past literature, and 3) our own results. We designed constructs to minimize differences; base stacked constructs are structurally identical (only differing by identity of a single base) and control constructs are structurally similar with the insertion of a small ssDNA gap. With this design, we expect extrapolation biases to equally affect all of the base stacked constructs and likely also the non-stacking controls, preserving the stacking energy estimates. Previous literature largely supports this model as well. Pulling on duplexes has been suggested to extrapolate to the equilibrium off-rate [57], even with variations in construct design, suggesting that the 1D model is a reasonable approximation. Further, the transition state only weakly depends on duplex length [36], and helicity and base pair tilt have been shown to not play a major role in duplex shearing [56]. On-rate experiments have shown consistent results between stacked and unstacked configurations similar to ours [27], and have been reported to be sequence independent for stacking interactions [26]. Together this body of evidence suggests that small structural variations do not alter the transition state appreciably. Our own experimental data in Figure 3 supports this empirically by exhibiting parallel slopes in the force dependence of different constructs, as well as transition state positions that match well with previous literature using different construct designs [57]. The close agreement of our non-equilibrium data with equilibrium measurements under force by Rieu et. al [27] also suggests our approach is valid.”

2. Regarding the fact that some of the curves aren't all that well fit by single exponentials: I'm not convinced by the authors' claim that it's just an artifact of small datasets. Many cases do indeed seem to be just that (e.g. Fig. 2g, some of Fig. S4, some of Fig. S8). However, in other cases there is a systematic effect where the initial drop-off is faster than the single-exponential fit and the later drop-off is slower even when there are well over 100 molecules (e.g. in Fig. 3b A|T at 15 and 20 pN, in Fig. 4 C|C and C|T blue curves, in Fig. S4 all replicates at 15 pN, in Fig. S7 A|T replicate 2, in much of Fig. S8). Some of these cases have ~500 molecules. I appreciate that of course there will be noise, but one would expect noise to be less systematic! Despite the good R^2 values, I expect that applying something like the non-parametric Wald-Wolfowitz runs test would find that an appreciable number of these fits fail the test. The concern of course is that some systematic error is thereby introduced into the analysis of the stacking energies because the rates are not correct.

One possible explanation is that the slow tail represents multiple tethers. That would require a more sizable fraction of the tethers to be multi-tethers than one would expect, given what the authors did to try to minimize multiple tethers, but it could account for the inconsistencies in the appearance of the effect (e.g. only one of the replicates at 15 pN in Fig. S3, with other 500 molecules, shows the non-single-exponential decay). It might be quite difficult to determine the source of this effect, but the authors should at least acknowledge it as a possible bias in the discussion section.

We have no problem acknowledging this possibility of bias – multiple tethers are always a concern and may have led to these minor imperfections in the data. We have added the following text:

“In some cases it can be observed that there is a continued slow decay over a time scale longer than the single exponential, which is also possibly from rare multiple tethers. While such non-idealities appear minor in our data, we acknowledge that these could provide a source of possible bias in our results.”

3. I think the authors have explained more clearly why their work represents an important advance over previous results. Nevertheless, the authors’ insistence on omitting a mention of the work by Rieu et al. measuring base stacking energies in the introduction when outlining the previous efforts at investigating the problem the authors are addressing is not acceptable from the point of view of scholarship. The authors’ contention that the paper by Rieu et al. was not published when the work was done and hence was not part of the background of the work is irrelevant: the authors are publishing their own work now, not two years ago (prior to publication of Rieu et al.), hence the paper by Rieu et al. is part of the existing scientific literature that must be taken into account when describing the context/background of what is known about base stacking and how it has been investigated. It would be very easy to include a mention of this reference in either paragraph 2 or possibly paragraph 3, pointing out that it introduced a promising method to measuring single-base stacking but did not apply it to systematically characterize stacking energies.

We will concede this point and added this text to the introduction in paragraph 2:

“One recent exception determined the energy of an individual A|G base stack as an application of a novel imaging technique with magnetic tweezers, but did not systematically investigate base stacking energies [27].”

4. The authors say that they don’t think it is constructive to explain the reasons why previous measurements have underestimated stacking energies. I disagree—in addition to discussing some of the features of the current measurement that might bias the results, it’s valuable to mention some of the features that might bias the previous experiments. I doubt that those factors have been discussed much in the literature, since disagreement with other results did not exist when those papers were published. The discussion does not need to be extensive, but I think it will be a helpful contribution to understanding the significance of this paper.

We agree. It is constructive - we are just concerned with highlighting potential limitations of other people’s work. We have added and revised text (underlined) integrating portions of our 1st response to reviewer into the discussion.

“Previous approaches have required several assumptions to arrive at their results. Deriving energies using stacked/unstacked equilibrium from migration of nicked DNA in urea gels [24] required assumptions about the effect of urea denaturant, the effect of physical forces in gel electrophoresis, and the effect of bending energy on kinked DNA. Deriving energies from measurements of single-molecule in end-stacking of DNA origami tubes [26] required assuming perfect formation and stability of complex DNA origami structures with sharp bends at the measured interfaces, a passive linking tether, and aligned contact across multiple interfaces with force shared evenly among them. It is worth pointing out that failure of most of these assumptions would be expected to result in measuring a base stacking value weaker than the actual value. Notably, one recent paper published with a similar construct design and relatively direct measurement approach found a single A|G base stack energy of -2 kcal/mol [27], consistent within error to our measurement.”

5. Something I did not notice previously: the grey and blue curves in Fig. 4 do not seem to be defined in the figure caption, this should be corrected.

Thanks, we added these definitions in the figure caption.